# Natural Identifiers for Privacy and Data Audits in Large Language Models

**Lorenzo Rossi, Bartłomiej Marek, Franziska Boenisch, Adam Dziedzic**[*]
CISPA Helmholtz Center for Information Security

## Abstract

Assessing the privacy of large language models (LLMs) presents significant challenges. In particular, most existing methods for auditing *differential privacy* require the insertion of specially crafted canary data *during training*, making them impractical for auditing already-trained models without costly retraining. Additionally, *dataset inference*, which audits whether a suspect dataset was used to train a model, is *infeasible* without access to a private non-member held-out dataset. Yet, such held-out datasets are often unavailable or difficult to construct for real-world cases since they have to be from the same distribution (IID) as the suspect data. These limitations severely hinder the ability to conduct scalable, *post-hoc* audits. To enable such audits, this work introduces **natural identifiers (NIDs)** as a novel solution to the above-mentioned challenges. NIDs are structured random strings, such as cryptographic hashes and shortened URLs, naturally occurring in common LLM training datasets. Their format enables the generation of unlimited additional random strings from the same distribution, which can act as alternative canaries for audits and as same-distribution held-out data for dataset inference. Our evaluation highlights that indeed, using NIDs, we can facilitate post-hoc differential privacy auditing *without any retraining* and enable dataset inference for any suspect dataset containing NIDs without the need for a private non-member held-out dataset.

## 1 Introduction

Large Language Models (LLMs) are increasingly used in applications like chatbots and text generation, where they are often trained on sensitive data, such as private conversations. Since LLMs have been shown to leak information about the training data (Carlini et al., 2019; 2021; Duan et al., 2024; Mattern et al., 2023), we need auditing methods to evaluate and quantify their privacy risks, ensuring safe deployment. Overall, there are two broad families of audits. *Formal* audits, *e.g.,* (Jagielski et al., 2020; Nasr et al., 2023; Panda et al., 2025; Steinke et al., 2023), aim to empirically verify claimed theoretical privacy guarantees of models trained with differential privacy (DP) (Dwork et al., 2006). *Standard* empirical privacy audits extend to models trained without privacy protection in mind and aim to understand the general leakage of individual training data points (Carlini et al., 2022; Duan et al., 2024; Shokri et al., 2017), or, in the case of dataset inference (DI) (Dziedzic et al., 2022; Maini et al., 2021; 2024), ask the question whether an entire data subset was used to train the model.

Unfortunately, both types of audits experience significant limitations in LLMs. One key limitation of the formal privacy auditing methods is that they require inserting canary data *during training*. As a result, these methods are inapplicable to pretrained LLMs without retraining, which is typically infeasible due to its high cost. Additionally, both types of audits rely internally on membership inference attacks (MIAs) (Shokri et al., 2017), where an adversary attempts to determine whether a particular data point was part of the model's training set. To be successful, MIAs require non-member held-out data from the exact same distribution as the member data used during training (Duan et al., 2024; Maini et al., 2024; Mattern et al., 2023; Shi et al., 2024). In practice, this data is usually hard to obtain, limiting the applicability of MIAs for audits. This limitation also equally affects DI, which assumes access to a held-out validation set that matches the distribution of the training data. Currently, the only widely used validation sets originate from the Pile (Gao et al., 2020), which is used in the

---

[*]For correspondence, please contact Franziska Boenisch (`boenisch@cispa.de`) and Adam Dziedzic (`dziedzic@cispa.de`).

training of Pythia models (Biderman et al., 2023), and to a lesser extent, the Dolma dataset (Soldaini et al., 2024), used in training the OLMo models (Groeneveld et al., 2024).

We identify *natural identifiers* (NIDs) as a solution to all the above-mentioned problems. NIDs are structured random strings, generated according to some well-defined criteria, such as outputs from secure hash algorithms (*e.g.,* MD5 or SHA-1), shortened URLs, or cryptocurrency wallet addresses. We observe that these strings are naturally included in datasets, such as discussion platforms (*e.g.,* StackExchange) and code repositories (*e.g.,* GitHub) that are used as part of the training corpora for state-of-the-art LLMs.[1] Especially code repositories are relevant for training powerful LLMs (Hui et al., 2024; Roziere et al., 2023) as, beyond supporting code generation, they also strengthen broader capabilities such as logical reasoning, problem solving, and world knowledge (Aryabumi et al., 2025; Petty et al., 2025; Kim et al., 2024; Hayase et al., 2024) which are important for LLMs' performance. **Our unique insight is that each of the popular NIDs has a known generation function that we can leverage to generate an *unlimited* number of held-out (non-member) data points from the same distribution as the NIDs, which are naturally included in real-world suspect sets.**

Equipped with these insights, we show how to leverage NIDs to perform formal post-hoc privacy auditing for LLMs. We build on the currently fastest single training run auditing approach (Steinke et al., 2023), which needs to include dedicated canaries prior to training. We demonstrate that when NIDs naturally occur in the training set, we can construct their corresponding auditing set *post-hoc* from the same distribution and retroactively assess the privacy guarantees of any LLM without the requirement of expensive retraining from scratch. Our privacy auditing with NIDs improves the lower bounds on the privacy parameters of an algorithm compared to the auditing framework by Steinke et al. (2023). It also significantly reduces the sample complexity, *i.e.,* it requires fewer NID canaries. Finally, in contrast to the one training run privacy auditing by Steinke et al. (2023), our method enables truly zero-run (*post-hoc*) audits of already pretrained LLMs.

Beyond formal audits, NIDs also make DI practically applicable, as one only has to identify NID types in the data subset that is suspected to be included in an LLM's training data, generate a held-out set consisting of NIDs of the same type, *i.e.,* from the same distribution, and then to perform the DI procedure (Maini et al., 2024). Thus, our fully post-hoc approach leverages NIDs to perform DI without any modifications to the training data, which contrasts with the prior approach by Zhang et al. (2024a) that requires injecting random canaries into the pretraining dataset. We empirically validate our approach in a controlled environment, using open-source LLMs and their known training data. Specifically, we use the Pythia suite of models with the Pile dataset and the OLMo model with the Dolma dataset. Our results show that we can accurately infer training membership across diverse data subsets without false positives, suggesting that our approach may be useful in real-world litigations (Coulter, 2024).

In summary, we make the following contributions:

1. We propose NIDs as a practical and scalable solution to a key challenge in LLM privacy research: conducting *post-hoc privacy audits* in real-world settings without requiring model retraining or access to a dedicated held-out set.

2. We adapt the one-run DP auditing framework (Steinke et al., 2023) to leverage NIDs, enabling truly post-hoc DP auditing of pretrained LLMs without modifying the training process and achieving tighter lower bounds.

3. We make DI more practical by creating the necessary held-out set post-hoc using the NIDs present in the suspect set and improving its efficiency by introducing a novel ranking-based test.

4. We conduct extensive empirical evaluations, demonstrating the effectiveness of our NIDs for post-hoc privacy assessment over multiple LLM families and training datasets.

---

[1]Indeed, we observe that the publicly available datasets used to train popular LLMs, such as the Pile (Gao et al., 2020) or Dolma (Soldaini et al., 2024), contain 30637 and 23571 different types of NIDs, respectively— showcasing the practical availability of NIDs. The large number of NID-types and new types constantly emerging makes it impossible to omit them through the web crawlers, thus NIDs are less prone to being excluded from the LLMs' training set.

## 2 BACKGROUND

**Differential Privacy (DP).** DP (Dwork et al., 2006) is a framework that limits privacy leakage by ensuring no individual's data significantly alters the outcome of a computation. A randomized mechanism $M$ satisfies $(\varepsilon, \delta)$-DP if, for any two inputs $x$ and $x'$ differing by one record and any measurable set $S$, the following holds, where $\varepsilon$ bounds leakage and $\delta$ is the failure probability:

$$P[M(x) \in S] \leq e^{\varepsilon} P[M(x') \in S] + \delta.$$

In this work we adopt the *under replacement* adjacency, where two datasets are considered neighbors if they differ only in the replacement of one candidate element (rather than by addition or removal).

**Auditing DP.** The goal of DP audits is to empirically estimate a lower bound on the privacy parameters $\varepsilon$ and $\delta$ post-training. These audits help evaluate the tightness of the theoretical analysis (Jagielski et al., 2020; Nasr et al., 2023) and can also reveal errors in the mathematical analysis or flaws in the algorithm's implementation (Tramer et al., 2022). Privacy auditing generally relies on retraining models and inserting canaries during training (Jagielski et al., 2020; Nasr et al., 2023; Steinke et al., 2023; Mahloujifar et al., 2025). While Steinke et al. (2023) reduce computational costs with a privacy auditing technique that only requires a single training run, for LLMs with trillions of parameters, even this can be prohibitively expensive. We build on their approach and leverage NIDs to remove the need for retraining altogether.

**Membership Inference Attacks (MIAs).** MIAs (Shokri et al., 2017) aim to determine whether a specific data point was included in a model's training set. They have diverse applications, and in this work, we focus on their use for privacy auditing (Steinke et al., 2023). While MIAs have been extensively explored for small-scale models, MIAs for LLMs are a much more challenging problem. The latest work (Duan et al., 2024; Maini et al., 2024; Zhang et al., 2024a) indicates that the success reported by previous MIAs on LLMs (Mattern et al., 2023; Shi et al., 2024) is rather due to a distribution shift than to the attacks' ability to distinguish between the member and non-member data points. A prominent example is the temporal distribution shift that occurs when data before a specific cutoff date is selected as members and data after the point is treated as non-members, resulting in differences in language, wording, or formatting styles. When evaluated in the correct setting without distribution shift, Maini et al. (2024) showed that most attacks do not outperform random guessing.

**Dataset Inference (DI).** DI (Maini et al., 2021) aims to resolve whether a given suspect dataset was used to train a model. While initially proposed for model ownership resolution (Maini et al., 2024; Dziedzic et al., 2022), DI was recently extended to identify training data in LLMs (Maini et al., 2024; Zhao et al., 2025). Beyond LLMs, DI has also been successfully applied to other types of generative models, including Diffusion Models (Dubiński et al., 2025) and Image Autoregressive Models (Kowalczuk et al., 2025). In general, DI extracts diverse training membership features for the individual data points in the suspect set using various MIAs, aggregates them, and applies statistical testing to reliably determine whether the suspect set was used to train the model.

**Limitations of DI.** DI's major limitation is that the method relies on access to a *private held-out set from the same distribution as a suspect set*. Prior work (Zhang et al., 2024a) argues that this makes DI inapplicable for real-world use-cases where such data is usually not available. As a solution, Zhang et al. (2024a) propose to inject random and meaningless canaries into the data and then test how the LLM ranks the selected canary among all alternatives. Since they assume access to the generator of the random canaries, they can provide the corresponding validation data points and avoid distribution shifts. The approach's reliance on inserted random strings reduces its practical applicability, as content creators would have to artificially include such specialized strings in their datasets and hide them from human readers. Additionally, web crawlers can be trained to omit such arbitrary context-free strings when scraping the data from the internet, reducing the likelihood of this data being included in LLMs' training data. Finally, this solution does not work for existing LLMs that were trained without the use of injected canaries. In contrast, our observation is that we can leverage NIDs that are naturally included in LLMs' training sets, mitigating the need to insert purely random strings and enabling auditing of existing pretrained LLMs without retraining. As an alternative solution to overcome DI's reliance on an IID held-out set, Zhao et al. (2025) proposed generating a synthetic held-out dataset by training a suffix-based generator on the suspect set, followed by a post-hoc calibration to reduce the distributional gap between the real and synthetic data. However, this approach is computationally expensive, requiring extensive training and calibration, and it still

results in a residual distributional shift between real and synthetic datasets. In contrast, our generated held-out set based on NIDs is from the exact same distribution as the suspect set.

## 3 NATURAL IDENTIFIERS (NIDS)

We introduce NIDs, explore their natural occurrence, and provide the intuition on how they address key challenges in LLM privacy research. We then present the notation and formalization of NIDs, which will serve as the foundation for the subsequent sections.

### 3.1 NIDS IN THE WILD

Conceptually, NIDs are structured random strings, generated according to some well-defined functions. Prominent examples include outputs from secure hash algorithms (*e.g.,* MD5 or SHA-1, SHA-256), shortened URLs, or cryptocurrency wallet addresses. Additionally, new types of NIDs, *e.g.,* produced through novel URL shortening approaches, are emerging continuously. Such strings are omnipresent on the internet, for example, in code repositories (*e.g.,* GitHub) and discussion platforms (*e.g.,* StackExchange). Since large parts of the data used to pretrain state-of-the-art LLMs are crawled from the internet, these NIDs get naturally included in the LLMs' training sets. We carefully extract the NIDs, as described in Appendix C.

While LLM providers may attempt to filter out natural NIDs during data crawling, auditors hold a structural advantage in this setting (Hönig et al., 2024; Radiya-Dixit et al.). Removing all natural NIDs is exceptionally challenging: even corpora with aggressive regex-based cleaning, URL canonicalization, PII filtering, and multistage deduplication, such as Dolma, still contain tens of thousands of distinct NID types, as detailed in Table 6 (Appendix D). For our approach, an auditor only needs to identify a small subset of NIDs in the suspect set to conduct effective post-hoc audits. This makes our approach robust even under strict data curation pipelines, thus making our solutions for LLM privacy auditing widely applicable.

We analyze a wide range of popular LLM training datasets, including Pile (Gao et al., 2020) and Dolma (Soldaini et al., 2024), and identify that all of them contain multiple types of NIDs with numerous examples per type. In Appendix D, we provide an overview of the analyzed subsets and contained NIDs in Table 6. Notably, datasets that include code snippets, such as StackExchange and GitHub, have a high number of NIDs. Additionally, large non-topic-specific corpora, such as RefinedWeb and Pile Common Crawl, also contain a significant number of NIDs. SHA-1 and MD5 are the most frequent types of NIDs overall. For some large subsets, such as RefinedWeb, we have as many as 16989 NIDs. For instance, Pile's entire validation and test set, which comprises approximately 0.2% of the entire Pile dataset, contains 293 NIDs. Furthermore, as shown in Table 6, even highly filtered and curated datasets such as Dolma (Soldaini et al., 2024) contain a substantial number of NIDs. This makes our solutions for LLM privacy auditing widely applicable.

### 3.2 LEVERAGING NIDS

What makes NIDs special is their rigorously specified format in combination with a sequence of random characters. Given that their format is known, it becomes possible to generate an *infinite number* of other random strings that follow the same distribution. In the following, we present the intuition on how this property contributes to solving the most pressing challenges in LLM privacy research, namely, the lack of IID held-out data.

**1) NIDs provide post-hoc DP audits.** We can use NIDs to perform post-hoc auditing for LLMs trained with DP. To do so, we build on the one-run privacy audit by Steinke et al. (2023). In their method, they select a set of canary data points to be included or excluded during a training run. After training, an auditor attempts to infer for each of these data points whether it was included or not. The fraction of correct guesses provides a lower bound on the DP parameters. Using our NIDs, retraining the model is no longer necessary. Instead, we generate random samples from the same distribution as the NIDs seen during training. The NIDs as natural canaries can be ranked against the generated ones, for auditing *without any retraining*, *i.e.,* truly post-hoc. Section 4 outlines our approach to using NIDs for post-hoc DP auditing.

**2) NIDs enable DI.** NIDs enable DI for suspect sets, *i.e.,* a dataset for which we want to assess whether it has been used to train a given LLM, without requiring a same-distribution private held-out set. As detailed above, DI relies on a private held-out set from the same distribution as the suspect set to perform its assessment—a requirement that is difficult to meet in practice. This is especially due to the challenge of obtaining same-distribution data post-hoc (Zhang et al., 2024a), rendering DI challenging or impractical. By generating large held-out sets from the same distribution, NIDs address this issue, thus enabling DI to detect if an LLM was trained on a suspect set. If the suspect set was part of the LLM's training data, it will react differently to the NIDs included in that set and their generated held-out counterparts. Otherwise, if it was not trained on the suspect set, its behavior will be the same over both sets, as both NIDs and their generated counterparts, since to the LLM, they will just be the same type of random strings. We detail the use of NIDs for DI in Section 5.

### 3.3 FORMALIZING NIDS

An *identifier (ID)* is produced by sampling randomness $z$ from a known distribution and applying a generator $W$, i.e., $v = W(z)$. The set of all possible IDs from this generator is $V = \{W(z) : z \in \mathcal{Z}\}$. A *Natural Identifier (NID)* is simply an ID that actually appears in a real dataset. Given such an NID, we can draw fresh random inputs $z'$ to generate additional IDs from the same distribution, which we call *Generated Identifiers (GIDs)*. Because the identifier space $V$ is extremely large, a newly generated GID is overwhelmingly unlikely to coincide with any existing NID in the data.

As a concrete example, consider `Ethereum` addresses. An Ethereum address is effectively a 160-bit identifier, obtained from a private key through a deterministic derivation process. Given an NID corresponding to an Ethereum address, we can use the associated generation function $W(z) := \texttt{ETH}(z)$ to generate new GIDs. In this case, the set $V$ is the set of all valid Ethereum addresses (see Appendix A for details on the structure of NIDs and GIDs, and Appendix B for examples). Additionally, the probability of generating a GID that exactly matches one of the NIDs in the training data is negligible, since the address space has size $2^{160} \approx 1.46 \times 10^{48}$.

The main property of NIDs is that a priori each ID $v \in V$ is equally likely to be generated and published because it only depends on the source of randomness. The second important property of NIDs is that they allow easy sampling from the set $V$. In the suspect datasets $D_{\text{sus}}$, which we are auditing, there are usually $m$ NIDs, with the corresponding sets $V_1, \ldots, V_m$. Although the underlying identifier space $V$ is extremely large, for computational purposes we restrict attention to a *finite candidate set*: for each detected NID $\hat{v}_i$, we sample $c - 1$ fresh GIDs and form $V_i = \{\hat{v}_i\} \cup \{c - 1 \text{ GIDs}\}$ with $|V_i| = c$. Furthermore, for each set $V_i$ where $i \in \{1, \ldots, m\}$, we denote the NID as $\hat{v}_i \in V_i$, and specifically, the NID that belongs to the suspect dataset as $\hat{v}_i \in D_{\text{sus}}$. Finally, we define $\Sigma_i$ as the set of all the permutations over $V_i$.

## 4 DP AUDITING WITH NATURAL IDENTIFIERS

Using our NIDs, we adapt the one-run DP auditing method proposed by Steinke et al. (2023) to create a novel post-hoc DP auditing. Their technique considers $m$ canary samples and uses coin flips to randomly determine which samples should be included in the training set. Therefore, it is a binary case of adding or removing a single sample (and selecting between two options) that requires further retraining. Subsequent works (Panda et al., 2025; Liu et al., 2025) build upon the settings and methods proposed in the original paper, thus requiring retraining. In our case, we differ from previous approaches by eliminating the need to retrain the model to insert canaries, since NIDs are inherently present in the data. Therefore, adding or removing multiple training examples independently is not required. This is particularly important for LLMs, for which retraining is prohibitively expensive and time-consuming. Furthermore, our method operates under more realistic assumptions compared to Kazmi et al. (2024), who, although they relax the assumption of retraining, require training a generative model that must then generate samples following the original training data distribution. Additionally, we do not strengthen the canary signal for the audit by surrounding the canaries with random tokens, as in Panda et al. (2025). Finally, compared with Mahloujifar et al. (2025), our method can be viewed as a ranking-based generalization, where the task is to correctly identify the true NID from a set of $c$ candidates, by requiring it to appear among the top-$r$ ranked positions, rather than only identifying it as the single top-1 candidate.

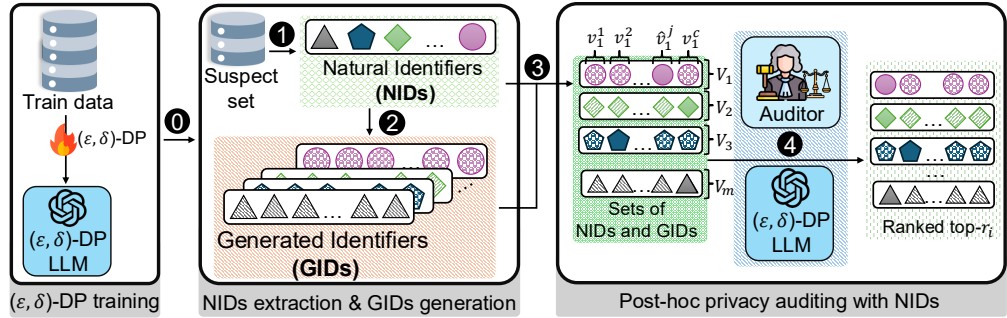

Figure 1: **Post-hoc DP auditing with NIDs and their corresponding GIDs.** ⓪ We consider the NIDs as the input to a training procedure $M$ (also referred to as the mechanism), which may satisfy $(\varepsilon, \delta)$-DP. ❶ Given a suspect dataset, we identify the NIDs. ❷ We generate the new $c - 1$ GIDs for each NID. ❸ We form the candidate sets $V_1, \cdots, V_m$ by combining the NIDs with corresponding GIDs. ❹ Given the resulting trained model and filtered NIDs with corresponding GIDs, an auditor seeks to infer, for each set $V_i$, which sample was the NID. To do so, the auditor ranks the samples in $V_i$ from the most to the least likely NID-candidate. A prediction is considered correct if the true NID appears among the top-$r_i$ ranked samples, where $r_i$ is a predefined threshold.

We show in Figure 1 how to leverage the NIDs to audit DP post-hoc. By leveraging the NIDs, our framework enables us to compute lower bounds on the privacy parameters of an algorithm without any additional training run of that algorithm. We first identify the NIDs that were present in the training data and denote their total number as $m$. For each NID $i \in \{1, \cdots, m\}$, we generate the corresponding GIDs, and the corresponding set of IDs $V_i = \{v_i^1, v_i^2, \ldots, \hat{v}_i^j, \ldots, v_i^c\}$, where we have $c - 1$ GIDs and a single NID denoted as $\hat{v}_i^j$. One of the main properties of NIDs is that, a priori, any element in $V_i$ could have been part of the training data in place of the NID. This enables us to model privacy auditing analogously to the fixed-length dataset variant proposed by Steinke et al. (2023). The key distinction in our approach is that, rather than selecting between two alternatives prior to training, we consider the NIDs as inserted canaries with the GIDs as multiple left-out canary possibilities for each set $V_i$. For this reason, the attacker's goal is to predict which sample was the NID by ranking the samples from the most likely to the least likely to be part of the training data. This offers more flexibility by enabling the attacker to represent uncertainty through a ranked list, rather than having to make a binary, top-1 inclusion decision.

Following the analysis of Theorem 5.2 by Steinke et al. (2023), we adapt their privacy auditing procedure to our setting to audit $(\varepsilon, \delta)$-DP mechanisms. We compare the rank of the real and alternative samples. For simplicity and clarity, we state the $\varepsilon$-DP version of the theorem, and in Appendix E, we show the complete theorem (Theorem 2) for the $(\varepsilon, \delta)$-DP case.

**Theorem 1** *Let $M : V_1 \times \cdots \times V_m \to \Sigma_1 \times \cdots \times \Sigma_m$ be an $\varepsilon$-DP mechanism under replacement. Let $S \in V_1 \times \cdots \times V_m$ be uniformly random, and define $T = M(S) \in \Sigma_1 \times \cdots \times \Sigma_m$. Then, for all $v \in \mathbb{R}$, all $t \in \Sigma_1 \times \cdots \times \Sigma_m$ in the support of $T$, all $r_1, \cdots, r_m$ with $r_i \leq |V_i|$, and $\frac{r_i e^{\varepsilon}}{|V_i| - 1 + e^{\varepsilon}} \leq 1$,*

$$\mathbb{P}_{\substack{S \leftarrow V_1 \times \cdots \times V_m, \\ T = M(S)}} [\sum_{i=1}^{m} \mathbb{1}[\mathrm{rank}(t_i, S_i) \leq r_i] \geq v | T = t]$$

$$\leq \mathbb{P}_{\hat{S} \leftarrow \mathrm{Bernoulli}(\frac{r_i e^{\varepsilon}}{|V_i| - 1 + e^{\varepsilon}})_{i=1}^{m}} [\hat{S} \geq v] := \beta(\varepsilon, v, t)$$

$\mathrm{rank}(a, b)$ *returns the 1-based position of the element $b$ in permutation $a$.*

In our setting, Theorem 1 states that if the mechanism (also referred to as the training procedure) is $\varepsilon$-DP, any attacker attempting to detect the NID is constrained. Concretely, the attacker ranks the mechanism's output on both the NID and its corresponding GIDs from most to least likely to be part of the training data without knowing which one is the NID. Then, they count how many NIDs appear in the top-$r$, where $r$ is a predefined threshold. The theorem states that this count is bounded by a Bernoulli distribution, whose probability depends on $\varepsilon$, $r$, and the number of GIDs.

Furthermore, compared to Theorem 5.2 by Steinke et al. (2023), Theorem 1 and Theorem 2 (presented in Appendix E) leverage a key property of NIDs: the ability to generate an unlimited number of GIDs (non-members).

Both theorems enable DP auditing through a hypothesis-testing framework. Moreover, in both cases, we can construct a confidence interval for a lower bound on $\varepsilon$. The proofs of Theorem 1 and Theorem 2 are provided in Appendix E.

**An Example of Our Privacy Auditing for the Randomized Response.** To illustrate our auditing framework, we use the classical randomized response mechanism (Warner, 1965). In this setting, each private value can either be revealed truthfully or replaced at random, with probabilities chosen to ensure $\varepsilon$-DP (see Appendix G.1 for the detailed description of the setting). The analogy to our framework is straightforward: each true value corresponds to an NID, and the alternative possibilities correspond to GIDs. The auditor ranks possible values given the output, and without any additional information, the best strategy is to place the observed output first. This yields a correct-guess probability matching the theoretical bound in Theorem 1. Figure 2 shows the empirical behavior of our auditor on randomized response for different set cardinalities $c = |V_i|$. We see that higher cardinality (i.e., more generated GIDs) is especially beneficial at larger privacy budgets ($\varepsilon \geq 8$), which is the typical regime in LLM training with DP (Duan et al., 2023; Li et al., 2022; Rossi et al., 2024; Hanke et al., 2024). This demonstrates how our framework scales naturally with the number of GIDs. Additionally, in Appendix F, we analyze the relationship between the number of samples $m$ (*i.e.,* number of NIDs) and $c$, as well as why a larger cardinality helps reduce the number of required samples.

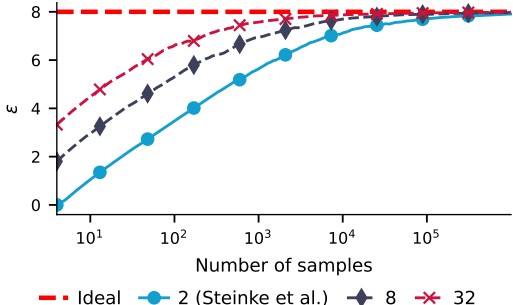

Figure 2: **Randomized response** with $\varepsilon = 8$ for different cardinalities $c = \{2, 8, 32\}$.

**Post-hoc DP Auditing Without Retraining in LLMs.** We verify that our proposed framework applies to privacy auditing in LLMs by adapting the black-box procedures proposed by Steinke et al. (2023) to the fixed-size dataset variant. The auditing process follows the algorithm described in Appendix H.3. Due to the lack of open-source private pretrained LLMs, to show the capabilities of our method, we finetune multiple Pythia models (70m, 160m, 410m, and 1b) using DP-SGD (Abadi et al., 2016) We use all NIDs extracted from the Pile test set (Gao et al., 2020). All lower and upper bounds are presented with 95% confidence intervals.

**Setup.** The training data consists $m = 197$ NIDs from the Github Pile test set, ensuring complete coverage of our assumption. Then, for each NID, we generate $c-1$ GIDs. In this way, we have sets of IDs $V_1, \ldots, V_m$. We set $\delta = 10^{-4}$ for various values of $\varepsilon$ using the Privacy Random Variable (PRV) accountant (Gopi et al., 2021), and finetune each model for 20 epochs using a maximum sequence length of 64 tokens and a clipping norm of 0.1. To rank each set of ID from most to least likely to be in the training data, we use Min-K% (Shi et al., 2024) and Loss (Yeom et al., 2018), and report the best result. By default, we set the ranking threshold to $r_i = 1$ (top-1) for all $i \in \{1, \ldots, m\}$. In this setting, a prediction is counted as correct only when the attacker's highest-scoring candidate coincides with the true NID. Complementary results for additional models and for thresholds $r_i > 1$ are reported in Appendix H.1.

**Higher Cardinality Improves Audits.** As a reference, we use the auditing of fixed-length datasets introduced by Steinke et al. (2023), which corresponds to a special case of our method where all sets $V_i$ have cardinality $c = 2$ and the corresponding threshold is $r_i = 1$. The empirical analysis in Figure 3 demonstrates that our method outperforms the baseline across multiple cardinality parameters ($c \in \{8, 32\}$) in fixed-length dataset settings. See Appendix H.1 for the results of the other models and for additional experiments with thresholds $r_i > 1$. Although higher cardinality can enhance the statistical power of the auditing procedure in the best-case scenario, meaning that fewer samples are required, the ranking task becomes increasingly complex. Instead of merely comparing two candidates, one must select from $c = |V_i|$ options. For smaller privacy budgets (*i.e.,* a more challenging prediction task), smaller cardinalities are beneficial. In contrast, for larger $\varepsilon$, higher cardinality tends to be advantageous and significantly outperforms the baseline. This trend aligns

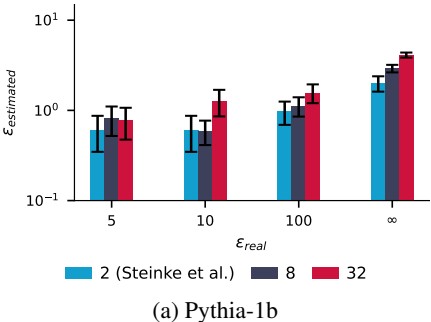
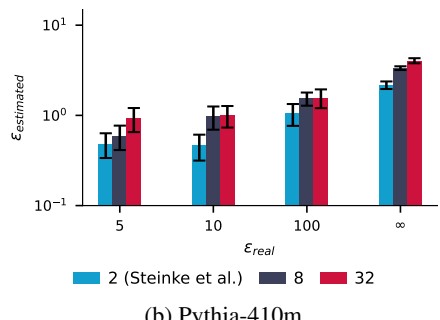

(a) Pythia-1b

(b) Pythia-410m

Figure 3: **Impact of cardinality ($c = \{2, 8, 32\}$) on $\varepsilon$ estimation**. Experiments conducted using $\varepsilon$ values of $\{5, 10, 100, \infty\}$. The case $c = 2$ corresponds to the method proposed by Steinke et al. (2023). The error bars represent a 95% confidence interval.

with our insights for randomized response, where increasing cardinality makes the privacy auditing more precise and tighter, particularly in less restrictive privacy settings.

## 5 DATASET INFERENCE WITH NIDS

Next, we turn to exploring the use of NIDs and our generated same-distribution GIDs for performing DI (Maini et al., 2021). As discussed in Section 2, the strongest limitation of DI is its reliance on a private held-out dataset from the same distribution as the suspect dataset, *i.e.,* the dataset for which we want to assess whether it was included in the training of the given model. Such datasets are often not available in practical applications (Zhang et al., 2024a). We present how our NIDs can overcome this limitation and enable successful DI for suspect datasets that contain NIDs. We experiment with Pythia-2.8b, 6.9b, 12b (trained on the Pile), and OLMo-7B[2] (trained on Dolma) to cover a range of model sizes and families. For ethical reasons, we focus on open models with known training data where we can *verify the correctness* our evaluation w.r.t. to the ground truth training sets, which is impossible for proprietary models where we have no access to the true training data.

Table 1: **MIAs on NIDs for Pythia-12b.** The AUC for MIAs between the NIDs and the corresponding GIDs on various subsets of the Pile dataset.

| MIA | Full Pile | | GitHub | | StackExchange | | Average | |
|---|---|---|---|---|---|---|---|---|
| | Train | Test | Train | Test | Train | Test | Train | Test |
| Loss | 58.6 | 50.3 | **71.8** | 51.1 | 50.3 | 50.9 | 60.2 | 50.8 |
| Min-K% | 57.6 | 51.0 | 68.4 | 50.6 | 50.7 | 51.2 | 58.9 | 50.9 |
| Min-K%++ | 56.9 | **51.4** | 71.2 | 50.3 | **50.8** | **51.9** | 59.6 | **51.2** |
| ReCALL | 53.5 | 50.2 | 50.6 | 50.3 | 50.0 | 51.1 | 51.4 | 50.5 |
| ReCALL(Hinge) | 51.3 | 50.1 | 53.3 | 50.4 | 50.4 | 51.4 | 51.7 | 50.6 |
| Hinge | **58.7** | 50.5 | **71.8** | **51.5** | 50.4 | 50.5 | **60.3** | 50.8 |

**MIAs for DI.** DI for LLMs (Maini et al., 2024) aggregates the outputs of multiple MIAs to extract a strong signal from the suspect data. We follow this approach and extract the signal from the suspect set's NIDs as a form of natural canaries. Therefore, we use MIAs on NIDs as a stepping stone for LLM DI. In this setting, the attacker aims to distinguish NIDs from their corresponding GIDs. For the training set, NIDs are drawn from the training data, while for the test set, they are drawn from the test data. In both cases, GIDs are constructed from data that was not used during training, serving as held-out samples. For the test set evaluation, we expect the AUC to be close to random guessing. This serves as a sanity check to confirm that the GIDs and NIDs come from the same distribution, since neither is present in the training data. To mimic the DI setting, we generate $c = 127$ new GIDs for each NID, balancing computational cost and distribution quality. Using our identified NID suspect set and the respective generated GIDs held-out set, we analyze existing state-of-the-art MIAs for LLMs, namely Loss (Yeom et al., 2018), Min-K% (Shi et al., 2024), Min-K%++ (Zhang et al., 2024b), ReCaLL (Xie et al., 2024), and Hinge (Carlini et al., 2022) to obtain useful signals for DI. For most MIAs, performance on the test set is close to random guessing, as expected, confirming no distribution shift between the NID suspect set and the generated GID held-out set. Train-test

---

[2]https://huggingface.co/allenai/OLMo-7B-0424-hf

behavior is well-calibrated, with higher average AUC on the train set. Results for Pythia-12b appear in Table 1; Appendix I reports additional models (Pythia, OLMo-7B) and TPR@1% FPR.

**DI on NIDs.** Given a suspect set $D_{\text{sus}}$, we first need to identify and extract all the NIDs in the dataset. The extracted NIDs form the suspect subset $D'_{\text{sus}}$, which we use to perform the DI. Then, for every real NID in $D'_{\text{sus}}$, we generate 127 new GIDs with the same NID type and with the same structure to form the held-out set from the same distribution as $D'_{\text{sus}}$. With the signal from the MIAs above, following Maini et al. (2024), we extract the features from the suspect set and $D'_{\text{sus}}$ and our generated held-out set. Next, following the DI protocol, we need to learn the correlation between the features (the MIA scores) and their membership status. To learn this correlation, we train a gradient boosting trees classifier to distinguish between the two distributions. To use all the samples available, we train and score the samples using K-Fold, and we ensure that the generated samples derived from a real sample end up in the same fold. Finally, following Maini et al. (2024), we perform statistical testing and compute the p-values. Under the null hypothesis, which assumes that the NIDs in the suspect set are not part of the training data, the ranks of each NID relative to its corresponding GIDs should follow a uniform distribution. This means that if we order the NIDs based on their association with GIDs, their positions should be evenly distributed across the ranking scale. We apply the Kolmogorov-Smirnov (KS) test to test this assumption. If the KS test detects a significant deviation from uniformity, we reject the null hypothesis, suggesting that the NIDs may, in fact, be present in the training data. Small p-values ($< 0.01$) indicate that we can reject the null hypothesis, *i.e.,* we are confident that the model was trained on the suspect set. Large p-values ($\gg 0.01$) suggest inconclusiveness of the test, *i.e.,* we are not confident whether the model was trained on the suspect set.

**Practical DI with NIDs.** Using our generated held-out set with GIDs and the suspect set $D'_{\text{sus}}$ with NIDs, we perform DI on various models and data subsets. Our main results for DI are summarized in Table 2 and Table 3. Compared with Maini et al. (2024), who used 1000 samples, we take much smaller suspect sets $D'_{\text{sus}}$ with 100 real NIDs to simulate a realistic setup. For each subset, we generate a held-out set using the NIDs, and perform DI. Our method shows that for the suspect sets that were included in the training data, DI obtains low p-values ($< 0.01$) that allow us to reject the null hypothesis. This highlights that the suspects are correctly identified as training data. At the same time, for test data (denoted as Test), *i.e.,* datasets that were not used to train the given LLM, we observe high p-values that do not allow us to reject the null hypothesis. The sets are, hence, correctly not marked as training data (p-values $\gg 0.01$). We present further results on models of various sizes and with varying numbers of NIDs in the suspect set in Figure 7 of Appendix J. The results highlight that the more NIDs are available in $D'_{\text{sus}}$, the more reliable the DI. Overall, using NIDs and the generated held-out set, we observe no false positives, while correctly identifying all training subsets (true positives). This highlights NIDs' ability to enable practical DI.

Table 2: **P-values for DI on the Pile Dataset with 100 suspect samples.** We use a 0.01 p-value threshold. We reject the null for all training subsets ($p \leq 0.01$) and do not reject it for the test set ($p > 0.01$). All outcomes are correct (✓).

| Model | GH | SE | HN | CC | AX | PM | IRC | Full | GH (Test) | Full (Test) |
|---|---|---|---|---|---|---|---|---|---|---|
| Pythia 12B | 0.0031 ✓ | 0.0001 ✓ | 0.0001 ✓ | 0.0001 ✓ | 0.0001 ✓ | 0.0001 ✓ | 0.0001 ✓ | 0.0001 ✓ | 0.8182 ✓ | 0.2847 ✓ |
| Pythia 6.9B | 0.0001 ✓ | 0.0001 ✓ | 0.0001 ✓ | 0.0002 ✓ | 0.0001 ✓ | 0.0001 ✓ | 0.0001 ✓ | 0.0001 ✓ | 0.6139 ✓ | 0.0811 ✓ |
| Pythia 2.8B | 0.0001 ✓ | 0.0001 ✓ | 0.0001 ✓ | 0.0001 ✓ | 0.0001 ✓ | 0.0001 ✓ | 0.0001 ✓ | 0.0001 ✓ | 0.9632 ✓ | 0.0660 ✓ |

Notation: GH = GitHub, SE = StackExchange, HN = HackerNews, CC = Pile-CC, AX = ArXiv, PM = PubMedCentral, IRC = UbuntuIRC

Table 3: **P-values for DI on the Dolma Dataset with 100 suspect samples.** We use a 0.01 p-value threshold. We reject the null for all training subsets ($p \leq 0.01$) and do not reject it for the test set ($p > 0.01$). All outcomes are correct (✓).

| Model | OWM | PeS2o | RFW | AStack | MWika | AX | C4 | PP2 (Test) |
|---|---|---|---|---|---|---|---|---|
| OLMo 7B | 0.0001 ✓ | 0.0001 ✓ | 0.0003 ✓ | 0.0001 ✓ | 0.0002 ✓ | 0.0001 ✓ | 0.0001 ✓ | 0.8961 ✓ |

Notation: OWM = OpenWebMath, RFW = RefinedWeb, AStack = Algebraic Stack, MWika = MegaWika, AX = ArXiv, PP2 = Proof Pile 2

**Controlled Ablations.** We also perform controlled ablations to characterize further how NID-based DI behaves under different design choices. First, we compare our NIDs against **standard injected canaries**, *i.e.,*, canaries that do not naturally occur in the training data but must be manually added. Although injected canaries fall outside our post-hoc threat model, this controlled setting helps

contextualize the strength of the NID leakage relative to existing auditing methods. We detail the choice and design of these canaries in Appendix K.1. Our results in Table 16 show that NIDs achieve competitive DI performance, measured in p-values. Second, we evaluate the impact of the GIDs being carefully sampled from the **same distribution** of the NIDs. Remember that DI critically depends on the GID generator matching the NID distribution: misimplementations that change casing produce strong signals for both members and nonmembers, thereby inflating false positives. To quantify this impact, we design GID generations that mismatch the original NIDs to various degrees. We describe our experimental setup in Appendix K.2. Our results show that deviations in distribution between NIDs and GIDs lead to false positives, highlighting the importance of our approach to generating GIDs exactly from the same distribution as NIDs. Third, we evaluate the impact of **stronger MIAs** on DI performance. Specifically, we augment the baseline features with CAMIA (Chang et al., 2024) and SURP (Zhang & Wu, 2024). See Appendix K.3 for details. Our results, shown in Table 18, indicate that adding more powerful MIAs consistently improves DI results. These results suggest that ongoing advances in MIA techniques further improve our framework's results. Fourth, we quantify whether the **identifier structure matters**. We construct a synthetic string that follows the format of each NIDs to measure the impact of the identifier structure. In Appendix K.4, we detailed the experimental setup. Our findings suggest that longer or more structured formats, such as SHA-512 and Java Serialization strings, yield the strongest DI signals, although shorter formats, such as MD5, still produce highly significant results, as shown in Table 19. Finally, we assess the impact of **increasing the number of NIDs** on the results of DI in Appendix K.5. Our results in Table 20 suggest that increasing the number of NIDs in the suspect set monotonically decreases the p-value in DI, illustrating the expected gains in statistical power.

**Task-Specific NIDs.** In some smaller, task-specific datasets, standard NIDs might be less common. To make DI practical in these settings, new task-specific NIDs can be discovered. As a case study, we consider the GSM8K dataset (Cobbe et al., 2021), a math word-problem dataset without standard NIDs. To generate valid and indistinguishable GIDs for DI, we create task-specific NIDs by treating each problem as a numeric template: for example, in "Natalia sold 48/2 = «48/2=24»24 clips in May. Natalia sold 48+24 = «48+24=72»72 clips altogether in April and May. #### 72.", we replace 48 and all dependent quantities (such as 24 and 72) with variables, resample consistent numbers to obtain a new problem, and use these as NIDs and GIDs. In Appendix B, we provide some practical examples of NIDs and the corresponding GIDs. See Appendix K.6 for details on the experimental setup. To assess whether the resulting NIDs and GIDs are suitable for our framework, we finetune Pythia-1b on 100 such NIDs, and run DI. The results in Table 4 show that this new task-specific type of NIDs produces statistically significant evidence for DI, confirming its effectiveness in various settings.

Table 4: **P-values for DI on GSM8K.** P-values obtained by our DI test on the GSM8K dataset, illustrating the effectiveness of task-specific NIDs.

| Number of NIDs | 50 | 60 | 70 | 80 | 90 | 100 |
|---|---|---|---|---|---|---|
| P-Value | $8.43 \times 10^{-4}$ | $9.56 \times 10^{-5}$ | $3.35 \times 10^{-4}$ | $1.63 \times 10^{-5}$ | $2.12 \times 10^{-6}$ | $1.60 \times 10^{-6}$ |

## 6 DISCUSSION AND CONCLUSIONS

We introduce the concept of *natural identifiers* (NIDs) as a practical and scalable solution to a central challenge in LLM privacy research: enabling *truly post-hoc privacy auditing*, *i.e.,* auditing models after training without requiring retraining or access to dedicated held-out data. This directly addresses a key limitation of most existing approaches, which rely on costly retraining procedures or artificially constructed held-out sets. While we focus on leveraging NIDs within the language domain for models trained on datasets containing such identifiers, our analysis shows that NIDs are *pervasively present* in standard LLM pretraining corpora. Their structured and reproducible nature enables the generation of an *unlimited number of non-member samples* from the same distribution, which we use to construct effective post-hoc auditing sets. Building on the one-run auditing framework, we demonstrate that NIDs yield *tighter DP bounds* with reduced sample complexity. By extending the task from binary classification to *ranking-based inference*, our approach further improves the flexibility and statistical power of privacy attacks. Beyond formal auditing, NIDs also make *DI* practically feasible using only the suspect data, without requiring access to held-out sets. Our empirical evaluations on open-source LLMs validate the effectiveness and practicality of this approach. In summary, NIDs offer a principled, both practical and efficient foundation for *real-world post-hoc privacy auditing*, advancing the feasibility of scalable and responsible privacy assessments for modern language models.

## 7 ETHICS STATEMENT

This work develops post-hoc auditing methods for LLMs using NIDs, which raises dual-use concerns: the same techniques that help auditors and regulators assess training-data usage and privacy guarantees could, in principle, be misused to better locate training artifacts or strengthen reconstruction attempts against weakly protected models. We acknowledge this risk, and believe such tools should be deployed only in controlled settings. At the same time, we view this kind of research as necessary: without realistic auditing techniques, it is difficult to verify privacy claims, detect misuse of training data, or incentivize stronger protections such as robust DP training.

## ACKNOWLEDGEMENTS

Franziska Boenisch received funding from the European Research Council (ERC) under the European Union's Horizon Europe research and innovation programme (grant agreement No 101220235). Additionally, we would like to acknowledge our sponsors, who support our research with financial and in-kind contributions: OpenAI and G-Research. We also thank members of the SprintML group for their feedback. Responsibility for the content of this publication lies with the authors.

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

## A    STRUCTURE OF NIDS AND GIDS

To extract the MIA signal, we use NIDs and their corresponding GIDs together with the surrounding textual context. Examples are provided in Appendix B. For each NID and its context, we generate a GID by replacing the NID with a randomly generated string that matches the original format, including structural features and casing patterns. This ensures that there is no distribution shift between the NID and its generated GIDs by construction. Each resulting string, whether it contains a NID or a GID, is limited to a maximum of 256 tokens. This includes both the identifier and its surrounding context. Within this limit, the final 64 tokens are reserved as a fixed suffix, and the remaining tokens are used for the prefix and the identifier itself. We ensure that both NIDs and GIDs are included in full and never partially truncated. All MIA signals are computed using these context-augmented strings. We include surrounding context to enhance the MIA signal, as prior work (Shi et al., 2024; Zhang et al., 2024b; Xie et al., 2024) has shown that longer input sequences can improve attack effectiveness.

## B    EXAMPLES OF NIDS AND GIDS

In this section, we show a series of examples to represent common appearances of the NIDs. We bold the parts that differ between the NIDs and GIDs. As shown in these examples, to create a new held-out sample, we only replace the NID with a GID. From the boxes below, we observe that a priori both the NID and the corresponding GID are equally likely to be part of the training data.

---

NID for MD5 from RefinedWeb Dolma (NID: `34d42a69a258fa51222a2e94b4563007`)

For a future birthday party – fairy party favors. But I want to figure out a different fairy, not Disney...
**34d42a69a258fa51222a2e94b4563007**.jpg 300×300 pixels
A quick, easy project for the kids: playful, pom-pom covered trees.
Carrot & Apple Cinnamon Streusel Muffins | a cup of mascarpone
Strawberry Banana Muffins recipe
PaperVine: Got Kids? Make your own Dinosaur Fossils!
Use modeling clay and some plastic dinosaurs to create dinosaur fossils. Made this last night to test it out. Turned out pretty cool. Trying to see if this would work for a kids event at work. I think it will! You only need 1 oz. of modeling clay per fossil.

---

GID for MD5 from RefinedWeb Dolma (GID: `9659875b92ba8fa639ba476aedbb73b9`)

For a future birthday party – fairy party favors. But I want to figure out a different fairy, not Disney...
**9659875b92ba8fa639ba476aedbb73b9**.jpg 300×300 pixels
A quick, easy project for the kids: playful, pom-pom covered trees.
Carrot & Apple Cinnamon Streusel Muffins | a cup of mascarpone
Strawberry Banana Muffins recipe
PaperVine: Got Kids? Make your own Dinosaur Fossils!
Use modeling clay and some plastic dinosaurs to create dinosaur fossils. Made this last night to test it out. Turned out pretty cool. Trying to see if this would work for a kids event at work. I think it will! You only need 1 oz. of modeling clay per fossil.

---

NID for SHA-1 from the training set of Dolma PeS2o (NID: `fac437a7d35ecfd53600ff4dc667563dfb251d25`)

Data availability
COPRO-Seq and INSeq datasets are deposited at the European Nucleotide Archive (ENA) under study accession: PRJEB38095. Proteomic data are available in the Mas-sIVE database under project number: MSV000085341. COPRO-Seq analysis software can be accessed at https://gitlab.com/hibberdm/COPRO-Seq and INSeq analysis soft-ware at https://github.com/mengwu1002/Multi-taxon_ analysis_pipeline; a copy has been archived at swh:1:rev: **fac437a7d35ecfd53600ff4dc667563dfb251d25**.
Additional information Competing interests Jeffrey I Gordon: Co-founder of Matatu, Inc., a company characterizing the role of diet-by-microbiota interactions in animal health. A provisional patent on the MFAB technology has been submitted (Washington University, assignee; PCT Application PCT/US2020/042678). The other authors declare that no competing interests exist.

GID for SHA-1 from Dolma PeS2o (GID: `95dfcf6dfc09c310e64c6540ad0b10e86394b0`

Data availability
COPRO-Seq and INSeq datasets are deposited at the European Nucleotide Archive (ENA) under study accession: PRJEB38095. Proteomic data are available in the MassIVE database under project number: MSV000085341. COPRO-Seq analysis software can be accessed at https://gitlab.com/hibberdm/COPRO-Seq and INSeq analysis software at https://github.com/mengwu1002/Multi-taxon_ analysis_pipeline; a copy has been archived at swh:1:rev: **95dfcf6dfc09c310e64c6540ad0b10e86394b006**.
Additional information Competing interests Jeffrey I Gordon: Co-founder of Matatu, Inc., a company characterizing the role of diet-by-microbiota interactions in animal health. A provisional patent on the MFAB technology has been submitted (Washington University, assignee; PCT Application PCT/US2020/042678). The other authors declare that no competing interests exist.

NID for GSM8K

**Question**
Natalia sold clips to 48 of her friends in April, and then she sold half as many clips in May. How many clips did Natalia sell altogether in April and May?
**Answer**
Natalia sold 48/2 = «48/2=24»24 clips in May.
Natalia sold 48+24 = «48+24=72»72 clips altogether in April and May.
#### 72

GID for GSM8K

**Question**
Natalia sold clips to 46 of her friends in April, and then she sold half as many clips in May. How many clips did Natalia sell altogether in April and May?
**Answer**
Natalia sold 46/2 = «46/2=23»23 clips in May.
Natalia sold 46+23 = «46+23=69»69 clips altogether in April and May.
#### 69

## C  POST-HOC EXTRACTION OF NIDS

We describe how to extract *natural identifiers* (NIDs) robustly. First, we select a series of regular expressions to identify potential *natural identifiers*. Depending on the type of secret, there might be a high number of false positives, therefore, we need to further remove invalid samples. We achieve that by first removing duplicates and then running a blind baseline (Das et al., 2024; Zhang et al., 2024a) using the n-grams as features and different types of tabular classifiers, such as Naive Bayes classifier, Gradient Boosting Trees, and Logistic Regression. Via K-Fold, we compute the MIA score of each sample, then we compare the rank of the real sample with respect to the generated ones. If the rank of the generated sample is too low or too high, we discard that sample.

We follow this procedure to robustly filter invalid *natural identifiers*. For instance, strings with "0123456789" are unlikely to be random strings and are most likely false positives. Finally, we check that the final blind baseline performance at the end of the filtering procedure is close to random guessing.

Table 5 summarizes the NID format, structure, and entropy. Additionally, for each type of NID, we have a specific way to generate them to closely resemble the original sample.
**MD5.** We generate the samples uniformly using this condition `[a-fA-F0-9]{32}` following the sample casing.
**SHA-1.** We generate the samples uniformly using this condition `[a-fA-F0-9]{40}` following the same casing of the original sample.
**SHA-256.** We generate the samples uniformly using this condition `[a-fA-F0-9]{64}` following the same casing of the original sample.
**SHA-512.** We generate the samples uniformly using this condition `[a-fA-F0-9]{128}`

following the same casing of the original sample.

**Ethereum Address.** We generate the samples uniformly using this condition `0x[a-fA-F0-9]{40}`. We select and generate only samples using case sensitivity as a checksum (EIP-55: Mixed-case checksum address encoding).

**Java serialization.** All serializable Java classes have the `serialVersionUID` attribute, which is often equal to a random number, for instance, `private static final long serialVersionUID = 6146619729108124872L`.

Table 5: Summary of NID formats, alphabets, and entropy in bits.

| NID Type | Length | Alphabet | Entropy |
|---|---|---|---|
| MD5 | 32 hex | `[0-9a-fA-F]` | 128 |
| SHA-1 | 40 hex | `[0-9a-fA-F]` | 160 |
| SHA-256 | 64 hex | `[0-9a-fA-F]` | 256 |
| SHA-512 | 128 hex | `[0-9a-fA-F]` | 512 |
| Ethereum Address | 40 hex | `[0-9a-fA-F]` | 160 |
| Java Serialization | ~20 digits | `[0-9]` | 64 |

Although the overall computational cost for processing trillions of tokens is not negligible—approximately one week of processing on a 128-core server—several considerations are important. First, the current implementation has not been optimized, and substantial acceleration could be achieved with relatively modest engineering improvements. Second, the cost of computing each NID is only on the order of tens of milliseconds, making the per-instance evaluation highly efficient. Most importantly, this approach is considerably less expensive than retraining large models from scratch. For example, a single training run of Pythia-12b with a highly optimized implementation requires approximately 72,300 hours of GPU computation. In contrast, our method avoids this prohibitive expense while still providing meaningful insights. Finally, it is not necessary to process the entire dataset; robust estimates can be obtained by sampling a substantially smaller subset, which further reduces the computational burden.

Once the NIDs are extracted, the GPU cost is relatively small, as it consists of running the model inference once or twice, depending on the MIA used, for each identifier. All the GPU experiments were conducted on a Linux server equipped with NVIDIA A100 GPUs.

## D  DISTRIBUTION OF NATURAL IDENTIFIERS

Table 6 shows for each subset and type of NID the number of NIDs. We highlight that large subsets, such as Dolma RefinedWeb, have a significant number of NIDs.

## E  FURTHER THEORY AND PROOFS

First, we state a useful definition and Lemma by Steinke et al. (2023), and then use them to prove Theorem 1.

**Definition 1 (Stochastic Dominance)** *[Definition 4.8, Steinke et al. (2023)] Let $X, Y \in \mathbb{R}$ be random variables. We say $X$ is stochastically dominated by $Y$ if $\mathbb{P}[X > t] \leq \mathbb{P}[Y > t]$ for all $t \in \mathbb{R}$.*

**Lemma 1** *[Lemma 4.9, Steinke et al. (2023)] Suppose $X_1$ is stochastically dominated by $Y_1$. Suppose that, for all $x \in \mathbb{R}$, the conditional distribution $X_2|X_1 = x$ is stochastically dominated by $Y_2$. Assume that $Y_1$ and $Y_2$ are independent. Then, $X_1 + X_2$ is stochastically dominated by $Y_1 + Y_2$.*

Here, we have the proof of Theorem 1.

*Proof:* Our analysis is similar to Proposition 5.1 by Steinke et al. (2023).
Fix some $t \in \Sigma_1 \times \cdots \times \Sigma_m$, and $i \in \{1, \ldots, m\}$, $a \in V_i$, and $s_{<i} \in V_1 \times \cdots \times V_i$. Using Bayes'

Table 6: **Natural Identifiers in Different Datasets.** We present the number of various *natural identifiers* (here: SHA-1, MD5, SHA-256, Java Serialization, SHA-512, and Ethereum Address) in the analyzed datasets. The *Total Number* denotes the total number of *natural identifiers* in a given dataset.

| Dataset | Total Number | SHA-1 | MD5 | SHA-256 | Java Serialization | SHA-512 | Ethereum Address |
|---|---|---|---|---|---|---|---|
| dolma RefinedWeb | 16989 | 8098 | 6192 | 2130 | 42 | 110 | 417 |
| pile train github | 13182 | 5389 | 1938 | 4158 | 819 | 701 | 177 |
| pile train stackexchange | 9862 | 4850 | 3235 | 1200 | 348 | 121 | 108 |
| pile train pile cc | 3422 | 1078 | 2008 | 274 | 1 | 8 | 53 |
| dolma algebraic stack train | 2384 | 1264 | 464 | 612 | 1 | 28 | 15 |
| pile train hackernews | 2268 | 1340 | 821 | 93 | 0 | 7 | 7 |
| dolma openwebmath train | 2207 | 1212 | 727 | 221 | 1 | 20 | 26 |
| pile train ubuntuirc | 1056 | 618 | 340 | 88 | 0 | 9 | 1 |
| dolma c4 | 791 | 408 | 301 | 63 | 0 | 4 | 15 |
| dolma PeS2o | 435 | 235 | 174 | 11 | 0 | 1 | 14 |
| dolma MegaWika | 383 | 115 | 200 | 62 | 0 | 2 | 4 |
| dolma ArXiv | 332 | 239 | 58 | 21 | 0 | 2 | 12 |
| Pile test (all subsets) | 293 | 130 | 69 | 62 | 13 | 14 | 5 |
| pile train pubmedcentral | 225 | 66 | 152 | 7 | 0 | 0 | 0 |
| pile train ArXiv | 207 | 75 | 122 | 7 | 0 | 0 | 3 |
| pile test github | 197 | 80 | 36 | 52 | 13 | 12 | 4 |
| pile train wikipediaen | 85 | 15 | 66 | 3 | 0 | 1 | 0 |
| pile test stackexchange | 58 | 34 | 16 | 6 | 0 | 2 | 0 |
| openwebmath test | 46 | 19 | 20 | 6 | 0 | 1 | 0 |
| algebraic stack test | 39 | 28 | 4 | 7 | 0 | 0 | 0 |
| dolma wiki | 38 | 11 | 22 | 3 | 0 | 2 | 0 |
| pile test pile cc | 18 | 6 | 8 | 3 | 0 | 0 | 1 |
| pile train philpapers | 16 | 1 | 15 | 0 | 0 | 0 | 0 |
| pile train freelaw | 15 | 1 | 14 | 0 | 0 | 0 | 0 |
| pile test hackernews | 13 | 7 | 6 | 0 | 0 | 0 | 0 |
| dolma tulu flan | 10 | 0 | 9 | 1 | 0 | 0 | 0 |
| pile test ubuntuirc | 5 | 3 | 2 | 0 | 0 | 0 | 0 |
| pile train enronemails | 4 | 0 | 4 | 0 | 0 | 0 | 0 |
| pile test wikipediaen | 2 | 0 | 1 | 1 | 0 | 0 | 0 |
| dolma books | 2 | 0 | 2 | 0 | 0 | 0 | 0 |
| pile train gutenbergpg 19 | 1 | 0 | 1 | 0 | 0 | 0 | 0 |
| pile train pubmedabstracts | 1 | 0 | 1 | 0 | 0 | 0 | 0 |

law and $\varepsilon$-DP, we have

$$\mathbb{P}[S_i = a | M(S) = t, S_{<i} = s_{<i}]$$

$$= \frac{\mathbb{P}[M(S) = t | S_i = a, S_{<i} = s_{<i}] \mathbb{P}[S_i = a]}{\mathbb{P}[M(S) = t | S_{<i} = s_{<i}]}$$

$$= \frac{\mathbb{P}[M(S) = t | S_i = a, S_{<i} = s_{<i}] \frac{1}{|V_i|}}{\sum_{j=1}^{|V_i|} \mathbb{P}[M(S) = t | S_i = V_{i,j}, S_{<i} = s_{<i}] \mathbb{P}[S_i = V_{i,j}]}$$

$$= \frac{\mathbb{P}[M(S) = t | S_i = a, S_{<i} = s_{<i}] \frac{1}{|V_i|}}{\sum_{j=1}^{|V_i|} \mathbb{P}[M(S) = t | S_i = V_{i,j}, S_{<i} = s_{<i}] \frac{1}{|V_i|}}$$

$$= \frac{1}{1 + \sum_{j=1, V_{i,j} \neq a}^{|V_i|} \frac{\mathbb{P}[M(S)=t|S_i=V_{i,j}, S_{<i}=s_{<i}]}{\mathbb{P}[M(S)=t|S_i=a, S_{<i}=s_{<i}]}} \in \left[ \frac{1}{1 + (|V_i| - 1)e^{\varepsilon}}, \frac{e^{\varepsilon}}{|V_i| - 1 + e^{\varepsilon}} \right]$$

Additionally, we can observe that for all $i \in \{1, \ldots, m\}$, we have that $\mathbb{P}[\text{rank}(t_i, S_i) \leq r_i] = \sum_{j=1}^{r_i} \mathbb{P}[S_i = t_{i,j}]$. Therefore, we can bound

$$\mathbb{P}[\text{rank}(t_i, S_i) \leq r_i] = \sum_{j=1}^{r_i} \mathbb{P}[S_i = t_{i,j} | M(S) = t, S_{<i} = s_{<i}]$$

$$\frac{1}{1 + (|V_i| - 1)e^{\varepsilon}} \leq \mathbb{P}[S_i = t_i, j | M(S) = t, S_{<i} = s_{<i}] \leq \cdot \frac{e^{\varepsilon}}{|V_i| - 1 + e^{\varepsilon}}$$

$$\frac{r_i}{1 + (|V_i| - 1)e^{\varepsilon}} \leq \mathbb{P}[\text{rank}(t_i, S_i) \leq r_i | M(S) = t, S_{<i} = s_{<i}] \leq \frac{r_i e^{\varepsilon}}{|V_i| - 1 + e^{\varepsilon}}$$

$$\mathbb{P}[\text{rank}(t_i, S_i) \leq r_i | M(S) = t, S_{<i} = s_{<i}] \in \left[ \frac{r_i}{1 + (|V_i| - 1)e^\varepsilon}, \frac{r_i e^\varepsilon}{|V_i| - 1 + e^\varepsilon} \right]$$

Thus, $\mathbb{P}[\text{rank}(t_i, S_i) \leq r_i | M(S) = t, S_{<i} = s_{<i}] \leq \frac{r_i e^\varepsilon}{e^\varepsilon + |V_i| - 1}$. With that, we can prove the result by induction. We inductively assume that $W_{m-1} := \sum_{i=1}^{m-1} \mathbb{1}[\text{rank}(t_i, S_i) \leq r_i]$ is stochastically dominated by $\hat{W}$ which is Bernoulli$(\frac{r_i e^\varepsilon}{|V_i| - 1 + e^\varepsilon})^{m-1}$. As above, $\mathbb{1}[\text{rank}(t_i, S_i) \leq r_i]$ is stochastically dominated by Bernoulli$(\frac{r_m e^\varepsilon}{e^\varepsilon + |V_m| - 1})$. By Lemma 4.9 by Steinke et al. (2023), $W_m = W_{m-1} + \mathbb{1}[\text{rank}(t_m, S_m) \leq r_m]$ is stochastically dominated by Bernoulli$(\frac{r_i e^\varepsilon}{|V_i| - 1 + e^\varepsilon})^m_{i=1}$. $\qquad\square$

To show the case $(\varepsilon, \delta)$-DP, we will first state Lemma 5.6 by Steinke et al. (2023). Then following the analysis of Proposition 5.7 and Theorem 5.2 by Steinke et al. (2023), we prove Theorem 2.

**Lemma 2** *[Lemma 5.6, Steinke et al. (2023)] Let $P$ and $Q$ be probability distributions over $\mathcal{Y}$. Fix $\varepsilon, \delta \geq 0$. Suppose that, for all measurable $S \subseteq \mathcal{Y}$, we have*

$$P(S) \leq e^\varepsilon \cdot Q(S) + \delta \quad \text{and} \quad Q(S) \leq e^\varepsilon \cdot P(S) + \delta.$$

*Then there exists a randomized function $E_{P,Q} : \mathcal{Y} \to \{0, 1\}$ with the following properties.*

*Fix $p \in [0, 1]$ and suppose $X \sim$ Bernoulli$(p)$. If $X = 1$, sample $Y \sim P$; and, if $X = 0$, sample $Y \sim Q$. Then, for all $y \in \mathcal{Y}$, we have*

$$\mathbb{P}_{X \sim \text{Bernoulli}(p), Y \sim XP + (1-X)Q} \left[ X = 1 \wedge E_{P,Q}(Y) = 1 \mid Y = y \right] \leq \frac{p}{p + (1-p)e^{-\varepsilon}}.$$

*Furthermore,*

$$\mathbb{E}_{Y \sim P}[E_{P,Q}(Y)] \geq 1 - \delta \quad \text{and} \quad \mathbb{E}_{Y \sim Q}[E_{P,Q}(Y)] \leq \delta.$$

**Theorem 2** *Let $M : V_1 \times \cdots \times V_m \to \Sigma_1 \times \cdots \times \Sigma_m$ be an $(\varepsilon, \delta)$-DP mechanism under replacement. Let $S \in V_1 \times \cdots \times V_m$ be uniformly random. Let $T = M(S) \in \Sigma_1 \times \cdots \times \Sigma_m$. Then, for all $v \in \mathbb{R}$, all $t \in \Sigma_1 \times \cdots \times \Sigma_m$ in the support of $T$, all $r_1, \ldots, r_m$ with $r_i \leq |V_i|$, and $\frac{r_i e^\varepsilon}{|V_i| - 1 + e^\varepsilon} \leq 1$,*

$$\mathbb{P}_{S \leftarrow V_1 \times \cdots \times V_m, T = M(S)}\left[ \sum_{i=1}^m \mathbb{1}[\text{rank}(t_i, S_i) \leq r_i] \geq v | T = t \right]$$

$$\leq \beta + \alpha\delta \sum_{i=1}^m |V_i|$$

*where*

$$\beta = \mathbb{P}_{\hat{S}}[\hat{S} \geq v],$$

$$\alpha = \max\left( \frac{1}{i} \mathbb{P}_{\hat{S}}[\hat{S} \geq v - i] : i \in \{1, \ldots, m\} \right),$$

$$\hat{S} \leftarrow \text{Bernoulli}\left( \frac{r_i e^\varepsilon}{|V_i| - 1 + e^\varepsilon} \right)^m_{i=1}.$$

Theorem 2 shows the analogous result of Theorem 1 using $(\varepsilon, \delta)$-DP.

Now, we show the proof of Theorem 2.

*Proof:* Our analysis follows Proposition 5.7 and Theorem 5.2 by Steinke et al. (2023).

For $i \in \{0, \ldots, m\}$ and $s_{\leq i} \in V_1 \times \cdots \times V_i$, let $M(s_{\leq i})$ denote the distribution on $\Sigma_1 \times \cdots \times \Sigma_m$ obtained by conditioning $\bar{M}(S)$ on $S_{\leq i} = s_{\leq i}$. We can express this as a convex combination:

$$M(s_{\leq i}) = \sum_{s_{>i} \in V_i \times \cdots \times V_m} M(s_{\leq i}, s_{>i}) \cdot \mathbb{P}_{S_{>i} \leftarrow V_i \times \cdots \times V_m}[S_{>i} = s_{>i}].$$

Additionally, for all $i \in \{1, \ldots, m\}$, and $a \in V_i$, we define $\hat{M}(s_{\leq i}, a)$ as the distribution on $\Sigma_1 \times \cdots \times \Sigma_m$ obtained by conditioning on $S_{\leq i} = s_{\leq i}$ and $S_{i+1} \neq a$, as follows:

$$\hat{M}(s_{\leq i}, a) = \sum_{b \in V_i, a \neq b} \frac{1}{|V_i| - 1} M(s_{\leq i}, b).$$

We define $S \leftarrow V_1 \times \cdots \times V_m$ to represent uniform sampling over $V_1 \times \cdots \times V_m$. For all $i \in \{1, \ldots, m\}$, we have that the distributions $P$ and $Q$ on $\Sigma_1, \ldots, \Sigma_m$, and let $E_{P,Q} : \Sigma_1, \ldots, \Sigma_m \rightarrow \{0, 1\}$ be the randomized function given by Lemma 2 (using $p = \frac{1}{|V_i|}$). Specifically, all $s_{\leq i} \in V_1 \times \cdots \times V_i$, all $t \in \Sigma_1 \times \cdots \times \Sigma_m$, and all $a \in V_i$, we have

$$\mathbb{P}_{S \leftarrow V_1 \times \cdots \times V_m, T \leftarrow M(S), E}[S_i = a \wedge E_{M(s_{<i}, a), \hat{M}(s_{<i}, a)}(T) = 1 | S_{\leq i} = s_{\leq i}, T = t] \leq \frac{e^\varepsilon}{|V_i| - 1 + e^\varepsilon},$$

$$\mathbb{E}_{S \leftarrow V_1 \times \cdots \times V_m, T \leftarrow M(S), E}[E_{M(s_{<i}, a), \hat{M}(s_{<i}, a)}(T) | S_{\leq i} = (s_{<i}, a)] \geq 1 - \delta.$$

For simplicity, for all $i \in \{1, \ldots, m\}$, we define $E_{M(s_{<i}, V_i)}(y) := \prod_{a \in V_i} E_{M(S_{<i}, a), \hat{M}(S_{<i}, a)}(y)$

and, for $b \in V_i$, we have

$$\mathbb{E}_{S \leftarrow V_1 \times \cdots \times V_m, T \leftarrow M(S), E}[E_{M(s_{<i}, V_i)}(T) | S_{\leq i} = (s_{<i}, b)] \geq 1 - |V_i|\delta.$$

For all $a \in V_i$, let $j := \operatorname{rank}(t_i, a)$, where we use 1-based ranks: rank 1 corresponds to the highest-scoring element, rank 2 to the next, and so on. So we can rewrite

$$\mathbb{P}_{S \leftarrow V_1 \times \cdots \times V_m, T \leftarrow M(S), E}[S_i = a \wedge E_{M(s_{<i}, V_i)}(T) = 1 | S_{\leq i} = s_{\leq i}, T = t]$$
$$= \mathbb{P}_{S \leftarrow V_1 \times \cdots \times V_m, T \leftarrow M(S), E}[\operatorname{rank}(t_i, S_i) = j \wedge E_{M(s_{<i}, V_i)}(T) = 1 | S_{\leq i} = s_{\leq i}, T = t].$$

Note that there is a bijective relationship between $a$ and $j$. Therefore, we have that

$$\mathbb{P}_{S \leftarrow V_1 \times \cdots \times V_m, T \leftarrow M(S), E}[\operatorname{rank}(t_i, S_i) \leq r_i \wedge E_{M(s_{<i}, V_i)}(T) = 1 | S_{\leq i} = s_{\leq i}, T = t] \leq \frac{r_i e^\varepsilon}{|V_i| - 1 + e^\varepsilon}.$$

For $j \in \{1, \ldots, m\}$, $s \in V_i \times \cdots \times V_m$, and $t \in \Sigma_1 \times \cdots \times \Sigma_m$, define

$$\widetilde{W}_j(s, t) := \sum_{i < j} \mathbb{1}[\operatorname{rank}(t_i, S_i) \leq r_i] \cdot E_{M(s_{<i}, V_i)}(t) = \sum_{i < j} \mathbb{1}[\operatorname{rank}(t_i, S_i) \leq r_i \wedge E_{M(s_{<i}, V_i)}(t) = 1]$$
$$\hat{W}_j(t) = \sum_{i \in [j]} S_i(t),$$

where, for each $i \in \{1, \ldots, m\}$ independently, $S(t)_i \leftarrow \operatorname{Bernoulli}\left(\frac{r_i e^\varepsilon}{|V_i| - 1 + e^\varepsilon}\right)$

By induction and Lemma 1, for any $j \in \{1, \ldots, m\}$ and $t \in \Sigma_1 \times \cdots \times \Sigma_m$, the conditional distribution $(\widetilde{W}_m(S, t) | M(S) = t)$ where $S \leftarrow V_1 \times \cdots \times V_m$ is stochastically dominated by $\hat{W}_m(t)$.

For $s \in V_1 \times \cdots \times V_m$ and $t \in \Sigma_1 \times \cdots \times \Sigma_m$, define

$$F(s, t) := \sum_{i=1}^{m} \mathbb{1}\left[E_{M(s_{<i}, V_i)}(t) = 0\right],$$

so that

$$W_m(s, t) := \sum_{i=1}^{m} \mathbb{1}[\operatorname{rank}(t_i, S_i) \leq r_i] \leq \hat{W}_m(s, t) + F(s, t).$$

Since the conditional distribution $(W_m(S,t)|M(S) = t)$, where $S \leftarrow V_1 \times \cdots \times V_m$ is stochastically dominated by $W_m(t)$, $W_m$ is stochastically dominated by the convolution $\hat{W}_m(T) + F(S,T)$. Finally, $F(s,t)$ is supported on $\{0, 1, \ldots, m\}$ and

$$\mathbb{E}[F(s,t)] = \sum_{i=1}^{m} \mathbb{P}[E_{M(s_{<i},a),\hat{M}(s_{<i},a)}(T) = 0] \leq \delta \sum_{i=1}^{m} |V_i|.$$

Since $\hat{W}_m(T)$ does not depend on $S$, the input $S$ does not contribute to the dependence between $F(S,T)$ and $W_m(T)$, so we can elide this input in the statement, that is, $F(T) = F(S,T)$ for $S$ drawn from an appropriate distribution.

Given these constraints, we can formulate finding the optimal distribution $F(t)$ for a given $t \in \Sigma_1 \times \cdots \times \Sigma_m$ and $v \in \mathbb{R}$ as a linear program:

$$\text{maximize} \qquad \mathbb{P}_{\check{W},F}[\check{W}(t) + F(t) \geq v] - \sum_{i=0}^{m} \mathbb{P}[F(t) = i] \cdot \mathbb{P}[\check{W}(t) \geq v - i]$$

$$\text{subject to} \qquad \mathbb{E}_F[F(t)] = \sum_{i=0}^{m} \mathbb{P}_F[F(t) = i] \cdot i \leq \delta \sum_{i=1}^{m} |V_i|,$$

$$\sum_{i=0}^{m} \mathbb{P}_F[F(t) = i] = 1, \text{ and}$$

$$\mathbb{P}_F[F(t) = i] \geq 0 \quad \forall i \in \{0, 1, \ldots, m\},$$

where $\check{W}(t) := \sum_{i=1}^{m} \mathbb{1}[\text{rank}(t_i, S_i) \leq r_i]$ for $S_i \leftarrow \text{Bernoulli}\left(\frac{r_i e^{\varepsilon}}{|V_i| - 1 + e^{\varepsilon}}\right)^m$.

By strong duality, the linear program above has the same value as its dual:

$$\text{minimize} \qquad \alpha \cdot \delta \sum_{i=1}^{m} |V_i| + \beta$$

$$\text{subject to} \qquad \alpha \cdot i + \beta \geq \mathbb{P}_{\check{W}}[\check{W}(t) \geq v - i] \quad \forall i \in \{0, 1, \ldots, m\},$$

$$\alpha \geq 0.$$

Any feasible solution to the dual gives an upper bound on the primal. So, in particular, we can use the solution provided by

$$\beta = \mathbb{P}_{\check{W}^*}[\check{W}^* \geq v],$$

$$\alpha = \max\left(\{0\} \cup \left\{\frac{1}{i}\left(\mathbb{P}_{\check{W}^*}[\check{W}^* \geq v - i] - \beta\right) : i \in \{1, 2, \ldots, m\}\right\}\right),$$

where $\check{W}^*$ is a distribution on $\mathbb{R}$ that satisfies $\mathbb{P}_{\check{W}^*}[\check{W}^* \geq v - i] \geq \mathbb{P}_{\check{W}}[\check{W}(t) \geq v - i]$ for all $i \in \{0, 1, \ldots, m\}$ and all $t$ in the support of $T$. $\qquad \square$

## F  SAMPLE COMPLEXITY ANALYSIS ON THE CARDINALITY

A natural question is what advantage arises from increasing the cardinality $c = |V_i|$ (for simplicity we assume that all sets $V_i$ have the same cardinality). By Theorem 1, the probability that a mechanism produces a correct guess within the top-$r$ elements is stochastically dominated by a Bernoulli random variable with success probability

$$p = \frac{re^{\varepsilon}}{c - 1 + e^{\varepsilon}}, \qquad \text{with } p \leq 1.$$

Thus, if we consider $m$ independent guesses, the total number of correct guesses is stochastically dominated by a $\text{Binomial}(m, p)$ random variable.

Applying the Bernstein inequality to this binomial distribution yields the following tail bound: for any $\beta \in (0, 1)$,

$$\mathbb{P}\left[\frac{\hat{S}}{m} \geq p + \frac{1}{3m}\log\left(\frac{1}{\beta}\right) + \sqrt{\frac{1}{9m}\log^2\left(\frac{1}{\beta}\right) + \frac{2p(1-p)}{m}\log\left(\frac{1}{\beta}\right)}\right] \leq \beta,$$

where $\hat{S} \sim \mathrm{Binomial}(m, p)$ and $\hat{S}/m$ represents the empirical accuracy (i.e., the observed fraction of correct guesses).

On the right-hand side of this inequality, the first term $p$ corresponds to the expected accuracy, while the remaining terms form the concentration margin. Among these, the dominant contribution for large $m$ and $c$ is

$$\sqrt{\frac{2p(1-p)}{m}\log\left(\frac{1}{\beta}\right)}.$$

To understand the scaling, observe that for fixed $r$ and $\varepsilon$, we have

$$p = \Theta\left(\tfrac{1}{c}\right), \qquad p(1-p) = \Theta\left(\tfrac{1}{c}\right).$$

Consequently, the concentration margin decays as

$$\Theta\left(\sqrt{\tfrac{1}{mc}}\right).$$

This shows that the accuracy concentrates faster as the cardinality $c$ increases: compared to the binary case ($c = 2$), the deviation shrinks by a factor of $\sqrt{2/c}$. In other words, larger cardinalities yield tighter accuracy concentration bounds, providing a clear sample complexity improvement over the 1-out-of-2 setting.

# G  RANDOMIZED RESPONSE ANALYSIS

In the following subsections, we analyze in detail our novel auditing method using randomized response.

## G.1  RANDOMIZED RESPONSE FORMALIZATION

We now provide the complete derivation of the auditing bound for randomized response in our setting. Formally, we are given $m$ samples, each corresponding to a private integer $v_i \in \{1, \ldots, c\}$. The randomized response mechanism releases

$$y_i = \begin{cases} v_i & \text{with probability } \frac{1}{c} + \gamma, \\ a & \text{with probability } \frac{1}{c} - \frac{\gamma}{c-1}, \quad \forall a \neq v_i, \end{cases}$$

where $\gamma = \frac{e^\varepsilon - 1}{c\left(1 + \frac{e^\varepsilon}{c-1}\right)}$ ensures $\varepsilon$-DP.

The auditor ranks the $c$ possible values from most to least likely. Since $y_i$ is always the most likely input to produce itself, the optimal strategy is to rank $y_i$ first and order the remaining values randomly. The probability of a correct guess is therefore

$$\mathbb{P}[\text{correct}] = \frac{e^\varepsilon}{c - 1 + e^\varepsilon},$$

which exactly matches the bound of Theorem 1.

## G.2  ADDITIONAL RANDOMIZED RESPONSE EXPERIMENTS FOR $r > 1$

Figure 4a and Figure 4b show additional results for the randomized response setting. We highlight that our method is tight for rank threshold $r = 1$, and the higher the $\varepsilon$, the larger the improvement given by a larger cardinality $c$.

In the specific case of randomized response, $r > 1$ is not tight, as there is no further information to exploit, as the attacker's best response is to give the mechanism response as the first choice and

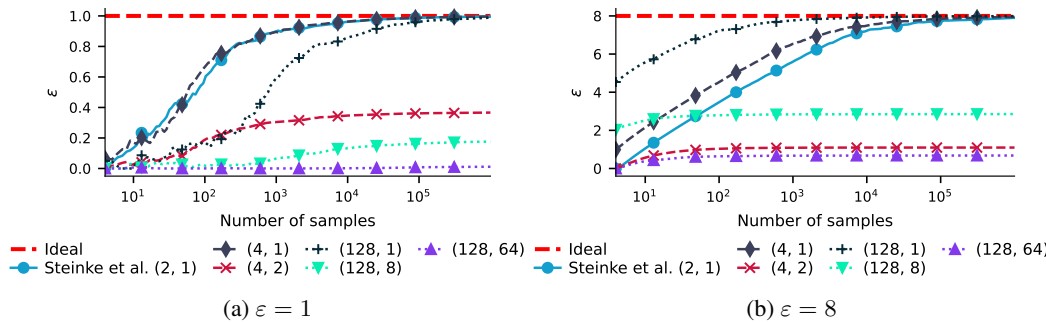

(a) $\varepsilon = 1$             (b) $\varepsilon = 8$

Figure 4: **Randomized response** mechanism with $\varepsilon = \{1, 8\}$. The red dashed line indicates the real $\varepsilon$ of the mechanism, while other ones indicate the estimated lower bound of $\varepsilon$ with 99% confidence for different choices of cardinality $c$, and rank threshold $r$. The (2,1) case corresponds to the method proposed by Steinke et al. (2023). Each label is written as (cardinality $c$, rank threshold $r$).

the other ones in random order. The randomized response mechanism returns a random value with a small bias towards the private one. From the auditor's point of view, the best attack returns the privatized value as the first option and the others in a random permutation. This means that the first value has some information about the private value, while the other ones have no information. Specifically, the probability of the first sample being the private sample is $\frac{e^{\varepsilon}}{c-1+e^{\varepsilon}}$, while for the other ones it is $\frac{1}{c-1+e^{\varepsilon}}$ (these results come from the randomized response output distribution). If we consider a certain threshold $r$, Theorem 1 roughly states that the probability of being correct is bounded by $\frac{re^{\varepsilon}}{c-1+e^{\varepsilon}}$. However, based on our attack, the probability that the correct value is in the top-$r$ is $\frac{e^{\varepsilon}}{c-1+e^{\varepsilon}} + \frac{r-1}{c-1+e^{\varepsilon}}$. For $r = 1$, we can observe that the two results match, while for $r > 1$, the attacker's probability is always strictly smaller than the ideal one (except for $\varepsilon = 0$). Theorem 1 and Theorem 2 give this additional flexibility of selecting the top-$r$ threshold; however, depending on the setting, this might be more or less useful.

## H    DP-SGD AUDITING

In the following subsections, we show additional experiments for DP-SGD auditing and the pseudocode of the auditing procedure.

### H.1    FURTHER EXPERIMENTS ON DP-SGD AUDITING

Figure 5 shows results for experiments conducted following settings described in Section 4 for other Pythia models (70m and 160m) (Biderman et al., 2023). The experiments substantiate observations from larger models, and the proposed framework constantly outperforms the baseline method proposed by Steinke et al. (2023).

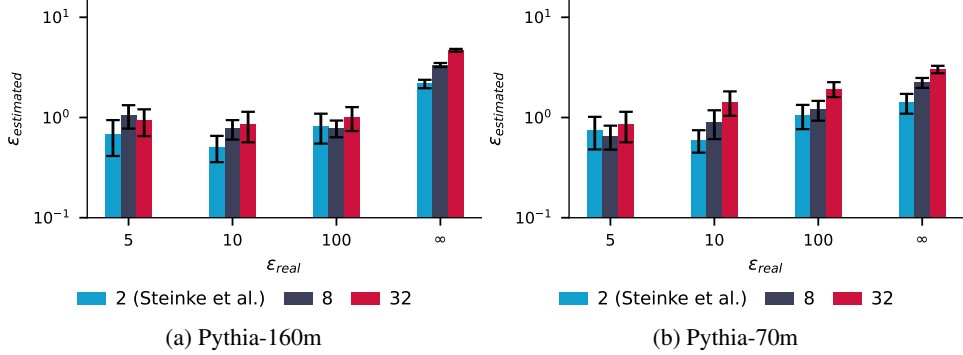

(a) Pythia-160m             (b) Pythia-70m

Figure 5: **Impact of cardinality ($c = \{2, 8, 32\}$) on $\varepsilon$ estimation** for other Pythia models.

Moreover, we explore different values of the $r$ parameter to estimate the lower bound of $\varepsilon$. The results shown in Figure 6 confirm our choice of parameter $r = 1$, thus providing the tightest and most reliable outcomes for our post-hoc DP auditing framework with NIDs.

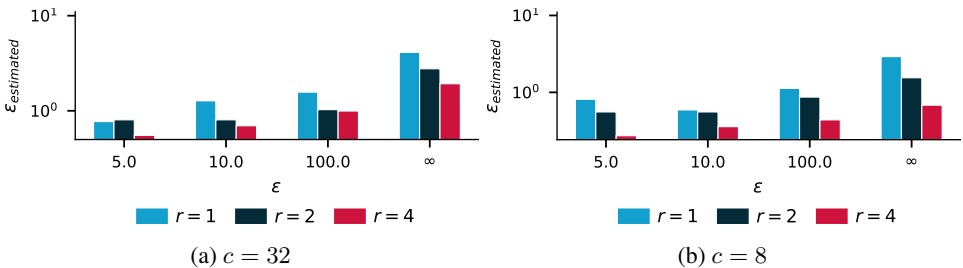

(a) $c = 32$            (b) $c = 8$

Figure 6: **Impact of rank** $r = \{1, 2, 4\}$ **on $\varepsilon$ estimation** for Pythia-1b.

## H.2   CONFIDENCE INTERVALS ACROSS 4 RANDOM SEEDS FOR DP AUDITING

Table 7 shows the confidence interval across 4 random seeds using Pythia-1b. The results show a low standard deviation across all of the settings.

Table 7: **DP auditing across 4 seeds.** Mean estimated $\varepsilon$ for Pythia-1b computed across 4 seeds.

| cardinality | $\varepsilon$ | Estimated $\varepsilon$ |
|---|---|---|
| 2 | 5 | $0.086 \pm 0.021$ |
| 2 | $\infty$ | $0.979 \pm 0.028$ |
| 2 | 10 | $0.106 \pm 0.020$ |
| 2 | 100 | $0.245 \pm 0.023$ |
| 8 | 5 | $0.720 \pm 0.144$ |
| 8 | 10 | $0.789 \pm 0.144$ |
| 8 | 100 | $1.094 \pm 0.129$ |
| 8 | $\infty$ | $2.329 \pm 0.058$ |
| 32 | 5 | $1.761 \pm 0.226$ |
| 32 | 10 | $1.830 \pm 0.231$ |
| 32 | 100 | $2.178 \pm 0.218$ |
| 32 | $\infty$ | $3.775 \pm 0.067$ |

## H.3   PSEUDOCODE FOR DP-SGD AUDITING

Algorithm 1 summarizes our approach for auditing DP-SGD using the results given by Theorem 2. We highlight that when for all $i \in \{1, \dots, m\}$, we have $|V_i| = 2$ and $r_i = 1$, the algorithm is equivalent to the fixed-length dataset case proposed by Steinke et al. (2023).

**Algorithm 1:** Adapted version of the black-box DP-SGD Auditor algorithm proposed by Steinke et al. (2023) for fixed-length dataset with NIDs.

**Input:** Dataset $D_0$, sets of canaries $V = \{V_1, \ldots, V_m\}$, the target ranks $r_1, \ldots, r_m$, and the DP-SGD settings

1: **for** $i \in \{1, \ldots, m\}$ **do**
2:     $S_i \leftarrow \text{Unif}\{V_i\}$
3: **end for**
4: $D_1 := \{V_{i,S_i} : i \in \{1, \ldots, m\}\}$
5: $D = D_0 \cup D_1$
6: Run DP-SGD on $D$ with given parameters, yielding $\{w^0, w^1, \ldots, w^\ell\}$
7: **for** $i \in \{1, \ldots, m\}$ **do**
8:     $Y_{i,j} \leftarrow \text{SCORE}(V_{i,j}; w^\ell) \quad \forall j \in [|V_i|]$
9:     $T_i \leftarrow \text{argsort}(Y_{i,j} \forall j \in [|V_i|])$
10: **end for**
11: $c \leftarrow 0$
12: **for** $i \in \{1, \ldots, m\}$ **do**
13:     **if** $T_{i,S_i} \leq r_i$ **then**
14:         $c \leftarrow c + 1$
15:     **end if**
16: **end for**
17: **return** Compute $\varepsilon_{\text{lower}}$ using the formula given by Theorem 2

## I    ADDITIONAL EVALUATION OF MIAS PERFORMANCE

Table 8, Table 10 and Table 11 show the MIA performance of the individual MIAs on the subsets of the Pile using the NIDs, where the goal is to distinguish the real from the generated ones. Furthermore, for completeness, we have Table 12, Table 13, Table 14, and Table 15 that show the MIA performance using TPR @ 1% FPR.

Table 8: **MIAs on NIDs for Pythia-12b.** The AUC for MIAs between the NIDs and the corresponding GIDs on various subsets of the Pile dataset.

| MIA | Full Pile | | Github | | Stack Exchange | | Ubuntu IRC | Wiki-pedia(en) | PubMed Central | Hacker News | Pile CC | ArXiv | Average | |
| | Train | Test | Train | Test | Train | Test | Train | Train | Train | Train | Train | Train | Train | Test |
|---|---|---|---|---|---|---|---|---|---|---|---|---|---|---|
| Loss | 58.6 | 50.3 | **71.8** | 51.1 | 50.3 | 50.9 | 50.3 | 50.6 | 50.6 | 60.5 | 51.1 | 50.4 | 54.9 | 50.7 |
| Min-K% | 57.6 | 51.0 | 68.4 | 50.6 | 50.7 | 51.2 | **51.1** | 50.6 | 50.7 | 60.5 | 52.3 | **51.0** | 54.8 | 50.9 |
| Min-K%++ | 56.9 | **51.4** | 71.2 | 50.3 | **50.8** | **51.9** | 51.1 | 51.3 | **51.1** | **69.7** | **53.2** | 50.9 | **56.2** | **51.2** |
| ReCALL | 53.5 | 50.2 | 50.6 | 50.3 | 50.0 | 51.1 | 50.3 | 51.3 | 50.2 | 57.8 | 50.1 | 50.2 | 51.6 | 50.5 |
| ReCALL(Hinge) | 51.3 | 50.1 | 53.3 | 50.4 | 50.4 | 51.4 | 50.5 | **51.9** | 50.8 | 50.3 | 50.4 | 50.0 | 51.0 | 50.6 |
| Hinge | **58.7** | 50.5 | 71.8 | **51.5** | 50.4 | 50.5 | 50.4 | 50.4 | 50.5 | 60.8 | 50.9 | 50.4 | 54.9 | 50.8 |

Table 9: **MIAs on NIDs for OLMo-7B.** The AUC for MIAs between the NIDs and the corresponding GIDs on various subsets of the Dolma dataset. All but Proof Pile 2 (Test) are part of the training data of Dolma.

| MIA | Dolma | | | | | | | | Average Train |
| | C4 | PeS2o | MegaWika | ArXiv | RefinedWeb | Algebraic Stack | OpenWebMath | Proof Pile 2 (Test) | Train |
|---|---|---|---|---|---|---|---|---|---|
| Loss | 50.1 | 50.2 | 50.2 | 51.2 | 50.1 | 50.0 | 50.9 | 50.6 | 50.4 |
| Min-K% | 50.1 | 50.2 | 50.5 | 51.3 | 50.1 | 50.5 | **51.7** | **51.3** | 50.6 |
| Min-K%++ | **50.4** | 50.2 | 50.0 | 50.7 | 50.1 | 50.2 | 50.8 | 51.0 | 50.3 |
| ReCALL | 50.2 | 50.9 | **51.0** | 50.7 | 50.1 | 50.4 | 51.0 | 51.0 | 50.6 |
| ReCALL (Hinge) | 50.3 | **51.4** | 50.2 | **51.9** | **50.2** | **50.7** | 50.2 | 51.0 | **50.7** |
| Hinge | 50.1 | 50.2 | 50.2 | 50.9 | 50.1 | 50.0 | 50.7 | 51.0 | 50.3 |

## J    FURTHER EXPERIMENTS ON DI

We evaluate DI on various models and data subsets. More concretely, we experiment with Pythia models 12b, 6.9b, and 2.8b and OLMo-7B. Additionally, we investigate the impact of increasing the number of samples in the suspect set. All results are summarized in Figure 7.

Table 10: **MIAs on NIDs for Pythia-6.9b.** The AUC for MIAs between the NIDs and the corresponding GIDs on various subsets of the Pile dataset.

| MIA | Full Pile | | Github | | StackExchange | | UbuntuIRC | Wikipediaen | PubMedCentral | HackerNews | Pile-CC | ArXiv | Average | |
|---|---|---|---|---|---|---|---|---|---|---|---|---|---|---|
| | Train | Test | Train | Test | Train | Test | Train | Train | Train | Train | Train | Train | Train | Test |
| Loss | 57.6 | 50.4 | **69.9** | 51.1 | 50.3 | 50.6 | 50.3 | 50.7 | 50.7 | 61.7 | 50.8 | 50.6 | 54.7 | 50.7 |
| Min-K% | 56.0 | 51.0 | 65.7 | 50.5 | 50.8 | **51.4** | 50.9 | 50.6 | 50.9 | 63.2 | 51.8 | 50.7 | 54.5 | 51.0 |
| Min-K%++ | 55.1 | 51.3 | 69.3 | 50.5 | **51.3** | 50.4 | **51.4** | **51.8** | **51.6** | **74.5** | **52.8** | **51.8** | **56.6** | 50.7 |
| ReCALL | 52.4 | **51.4** | 55.9 | 51.1 | 50.1 | 51.0 | 50.1 | 50.5 | 50.4 | 60.3 | 50.3 | 50.7 | 52.3 | **51.2** |
| ReCALL (Hinge) | 51.2 | 50.6 | 53.2 | 51.2 | 50.1 | 50.1 | 51.0 | 50.9 | 50.1 | 52.6 | 50.0 | 50.0 | 51.0 | 50.6 |
| Hinge | **57.7** | 50.7 | 69.9 | **51.6** | 50.4 | 50.1 | 50.2 | 50.0 | 50.7 | 61.7 | 50.7 | 50.3 | 54.6 | 50.8 |

Table 11: **MIAs on NIDs for Pythia-2.8b.** The AUC for MIAs between the NIDs and the corresponding GIDs on various subsets of the Pile dataset.

| MIA | Full Pile | | Github | | StackExchange | | UbuntuIRC | Wikipediaen | PubMedCentral | HackerNews | Pile-CC | ArXiv | Average | |
|---|---|---|---|---|---|---|---|---|---|---|---|---|---|---|
| | Train | Test | Train | Test | Train | Test | Train | Train | Train | Train | Train | Train | Train | Test |
| Loss | 52.8 | 50.0 | 58.9 | 50.4 | 50.2 | 50.2 | 50.1 | 50.5 | 50.5 | 60.3 | 50.8 | 50.6 | 52.8 | 50.2 |
| Min-K% | 52.1 | **52.4** | 59.5 | **52.9** | 50.6 | 50.3 | 50.2 | 50.1 | **50.6** | 61.6 | 51.6 | 50.5 | 53.0 | **51.8** |
| Min-K%++ | 50.3 | 52.3 | 58.2 | 50.6 | **50.9** | 50.1 | 50.2 | 50.2 | 50.3 | **73.6** | **52.8** | **51.4** | **54.2** | 51.0 |
| ReCALL | **53.7** | 51.1 | **64.4** | 52.2 | 50.1 | 50.1 | 50.2 | 50.8 | 50.5 | 58.0 | 50.2 | 51.1 | 53.2 | 51.2 |
| ReCALL (Hinge) | 50.9 | 50.6 | 50.9 | 50.8 | 50.5 | **50.7** | **50.9** | **52.3** | 50.2 | 51.3 | 50.2 | 50.1 | 50.8 | 50.7 |
| Hinge | 53.0 | 50.4 | 58.9 | 51.1 | 50.3 | 50.3 | 50.2 | 50.2 | 50.5 | 59.9 | 50.7 | 50.4 | 52.7 | 50.6 |

Table 12: **MIAs on NIDs for Pythia-12b.** The TPR @ 1% FPR for MIAs between the NIDs and the corresponding GIDs on various subsets of the Pile dataset.

| MIA | Full Pile | | Github | | StackExchange | | UbuntuIRC | Wikipediaen | PubMedCentral | HackerNews | Pile-CC | ArXiv | Average | |
|---|---|---|---|---|---|---|---|---|---|---|---|---|---|---|
| | Train | Test | Train | Test | Train | Test | Train | Train | Train | Train | Train | Train | Train | Test |
| Loss | 1.2 | 0.0 | 1.9 | 0.0 | 1.0 | 0.1 | 0.0 | 0.1 | 0.5 | 0.1 | 0.9 | 0.3 | 0.7 | 0.0 |
| Min-K% | 1.1 | 0.0 | 1.6 | 0.0 | 1.0 | **1.8** | 0.3 | 0.9 | 1.0 | 0.2 | 0.9 | 0.6 | 0.9 | 0.6 |
| Min-K%++ | **1.3** | 1.1 | **2.0** | 1.1 | 0.8 | 1.3 | 0.4 | 0.9 | 1.9 | 0.8 | 1.3 | 0.4 | 1.1 | 1.2 |
| ReCALL | 1.2 | 0.2 | 1.5 | 0.0 | 1.0 | 1.5 | **1.4** | 0.7 | 0.8 | 0.9 | **1.9** | 1.0 | 1.1 | 0.5 |
| ReCALL (Hinge) | 1.1 | **1.2** | 1.9 | **1.5** | 0.6 | 1.3 | 0.5 | **1.0** | 0.1 | 1.5 | 1.3 | **2.8** | 1.2 | 1.3 |
| Hinge | 0.0 | 0.4 | 0.0 | 0.5 | 0.9 | 1.5 | 0.5 | 0.5 | **2.1** | 1.1 | 0.9 | 1.3 | 0.8 | 0.8 |

Table 13: **MIAs on NIDs for Pythia-6.9b.** The TPR @ 1% FPR for MIAs between the NIDs and the corresponding GIDs on various subsets of the Pile dataset.

| MIA | Full Pile | | Github | | StackExchange | | UbuntuIRC | Wikipediaen | PubMedCentral | HackerNews | Pile-CC | ArXiv | Average | |
|---|---|---|---|---|---|---|---|---|---|---|---|---|---|---|
| | Train | Test | Train | Test | Train | Test | Train | Train | Train | Train | Train | Train | Train | Test |
| Loss | 1.2 | 0.1 | 1.9 | 0.0 | 1.0 | 1.3 | 0.4 | 0.0 | 0.3 | 0.3 | 0.5 | 1.0 | 0.7 | 0.5 |
| Min-K% | 1.1 | 0.1 | 1.6 | 0.0 | **1.3** | 0.7 | 0.5 | 1.0 | 0.9 | 0.3 | 1.0 | 1.3 | 1.0 | 0.3 |
| Min-K%++ | 0.9 | 0.7 | 1.0 | 0.6 | 1.2 | 1.4 | 0.4 | 1.3 | 0.9 | 0.9 | 0.4 | 1.1 | 0.9 | 0.9 |
| ReCALL | 1.0 | 0.2 | 1.5 | 0.0 | 1.2 | 1.3 | **1.1** | 0.6 | 1.2 | 1.2 | 1.2 | **2.1** | 1.2 | 0.5 |
| ReCALL (Hinge) | **1.3** | **1.4** | **2.0** | **1.5** | 0.5 | **2.6** | 0.6 | **2.3** | **1.8** | **3.3** | 1.2 | 1.8 | **1.6** | **1.9** |
| Hinge | 0.0 | 0.3 | 0.0 | 0.5 | 0.8 | 1.2 | 0.7 | 0.3 | 1.2 | 1.0 | 0.7 | 0.9 | 0.6 | 0.7 |

# K    CONTROLLED ABLATION OF DI

In this section, we investigate how the main design choices affect the behavior of our method. To carry out this controlled analysis, it is necessary to train a model for each configuration under study. Fully training a large model for every variation is computationally infeasible, and therefore, following the procedure described in Section 4, we finetune a smaller model that serves as a practical proxy for evaluating the influence of individual components. This controlled setup enables us to enforce the formatting rules of task-specific NIDs with precision, ensuring that the experiments isolate structural properties rather than reflecting irregularities present in real-world data. The following subsections present the corresponding evaluations conducted within this framework.

## K.1    COMPARISON WITH INJECTED CANARIES

In this subsection, we compare the performance of NIDs and commonly used injected canaries. Although injected canaries fall outside our post-hoc threat model, this controlled setting helps contextualize the strength of the NID leakage relative to existing auditing methods. In particular, we consider four types of canaries: (1) random alphabetic strings of length 32, (2) the NIDs (from the GitHub subset of the Pile test set), (3) fully IID strings (in-distribution text from the Pile test set), and (4) random hexadecimal strings of length 32. For each type, we inject 100 canaries into the training data and run DI. The resulting p-values for each canary type are reported in Table 16. Overall, we find that NIDs perform competitively with other injected canaries. They capture privacy

Table 14: **MIAs on NIDs for Pythia-2.8b.** The TPR @ 1% FPR for MIAs between the NIDs and the corresponding GIDs on various subsets of the Pile dataset.

| MIA | Full Pile | | Github | | StackExchange | | UbuntuIRC | Wikipediaen | PubMedCentral | HackerNews | Pile-CC | ArXiv | Average | |
|---|---|---|---|---|---|---|---|---|---|---|---|---|---|---|
| | Train | Test | Train | Test | Train | Test | Train | Train | Train | Train | Train | Train | Train | Test |
| Loss | 1.1 | 0.0 | 1.4 | 0.0 | 0.9 | 1.3 | 0.4 | 0.0 | 0.8 | 0.1 | 0.6 | 1.0 | 0.7 | 0.4 |
| Min-K% | 1.1 | 0.0 | 1.2 | 0.0 | **1.1** | 1.4 | 0.4 | **1.1** | 1.1 | 0.3 | 0.7 | 0.7 | 0.8 | 0.5 |
| Min-K%++ | 0.9 | 0.6 | 1.3 | 0.5 | 0.8 | 1.5 | 0.3 | 1.0 | **2.3** | 0.8 | 1.0 | 0.3 | 1.0 | 0.9 |
| ReCALL | 0.1 | 0.0 | 0.5 | 0.0 | 1.0 | 0.1 | 1.5 | 0.1 | 0.9 | 0.7 | **1.0** | **1.7** | 0.8 | 0.0 |
| ReCALL (Hinge) | **1.3** | **0.7** | **1.6** | **1.0** | 0.7 | 0.1 | **1.7** | 0.4 | 0.1 | **2.4** | 0.8 | 1.1 | 1.1 | 0.6 |
| Hinge | 0.1 | 0.4 | 0.1 | 0.4 | 0.8 | **1.5** | 0.4 | 0.2 | 1.5 | 1.2 | 0.9 | 0.9 | 0.7 | 0.8 |

Table 15: **MIAs on NIDs for OLMo 7B.** The TPR @ 1% FPR for MIAs between the NIDs and the corresponding GIDs on various subsets of the Dolma dataset.

| MIA | Dolma | | | | | | | | Average |
|---|---|---|---|---|---|---|---|---|---|
| | C4 | PeS2o | MegaWika | ArXiv | RefinedWeb | algebraic stack | openwebmath | Proof Pile 2 (Test) | Train |
| Loss | 0.4 | 0.9 | 0.4 | **1.2** | 0.8 | 0.9 | 0.3 | **0.0** | 0.7 |
| Min-K% | 0.7 | 0.5 | **1.5** | 0.3 | 0.9 | 0.5 | 0.4 | 0.0 | 0.7 |
| Min-K%++ | **1.1** | 0.8 | 0.2 | 0.8 | **2.0** | 0.3 | 0.9 | 0.0 | 0.9 |
| ReCALL | 0.7 | 0.6 | 0.6 | 0.7 | 0.7 | 0.9 | 0.6 | 0.0 | 0.7 |
| ReCALL (Hinge) | 0.7 | 0.3 | 1.1 | 0.2 | 0.2 | **1.1** | **2.2** | 0.0 | 0.8 |
| Hinge | 0.9 | **1.0** | 1.0 | 0.6 | 1.1 | 1.1 | 0.9 | 0.0 | 0.9 |

leakage more effectively than IID canaries and outperform random hexadecimal canaries, though some carefully crafted canaries, such as alphabetic strings, exhibit slightly stronger signals.

Table 16: **P-values for DI on Injected Canaries.** P-values obtained for each injected canary type.

| Canary Type | P-Value |
|---|---|
| Alphabetic | $< 1.00 \times 10^{-300}$ |
| NIDs (All subsets) | $4.17 \times 10^{-211}$ |
| NIDs (GitHub subset) | $3.31 \times 10^{-156}$ |
| IID | $4.55 \times 10^{-100}$ |
| Hex | $7.00 \times 10^{-23}$ |

### K.2 MISIMPLEMENTED GENERATOR

To study the benefits of our method for operating on held-out data from the same distribution, we analyze scenarios in which the held-out data are generated from a distribution that differs from the distribution of the suspect set data. We evaluate the impact of an incorrectly implemented generator. If the GID generator is not implemented properly, this induces a distributional shift between NIDs and GIDs. Starting from the correct generator, we construct three misimplemented variants that (1) flip the casing of alphabetic characters, (2) produce identifiers whose length is off by one, and (3) produce identifiers whose length is off by two. We then run DI on models finetuned on correct NIDs, but evaluated using imperfect GIDs, using the same protocol as in previous subsections. Table 17 reports the resulting p-values for member and non-member datasets. The results show that DI is sensitive to certain generator failures: for example, incorrect casing yields strong signals for both members and non-members, substantially inflating false positives. In contrast, modest length mismatches have a smaller impact on non-member p-values, likely because Min-K% and Min-K%++ only depend on the top-k tokens and are therefore relatively insensitive to appending or removing a small number of additional tokens. This analysis complements our microanalysis of NID formats and highlights that both structural differences and shifts in the identifier-generation distribution can meaningfully affect DI outcomes.

Table 17: **Misimplemented Generator.** P-values for DI on member and non-member datasets under different GID generator failures compared to the correct generator.

| Generator Failure | P-Value Members | P-Value Non-Members |
|---|---|---|
| Wrong Case | $< 1.00 \times 10^{-300}$ | $1.16 \times 10^{-54}$ |
| Length Off By 2 | $3.64 \times 10^{-99}$ | $5.39 \times 10^{-02}$ |
| Length Off By 1 | $< 1.00 \times 10^{-300}$ | $3.40 \times 10^{-01}$ |
| Correct Generator | $3.31 \times 10^{-156}$ | $9.83 \times 10^{-01}$ |

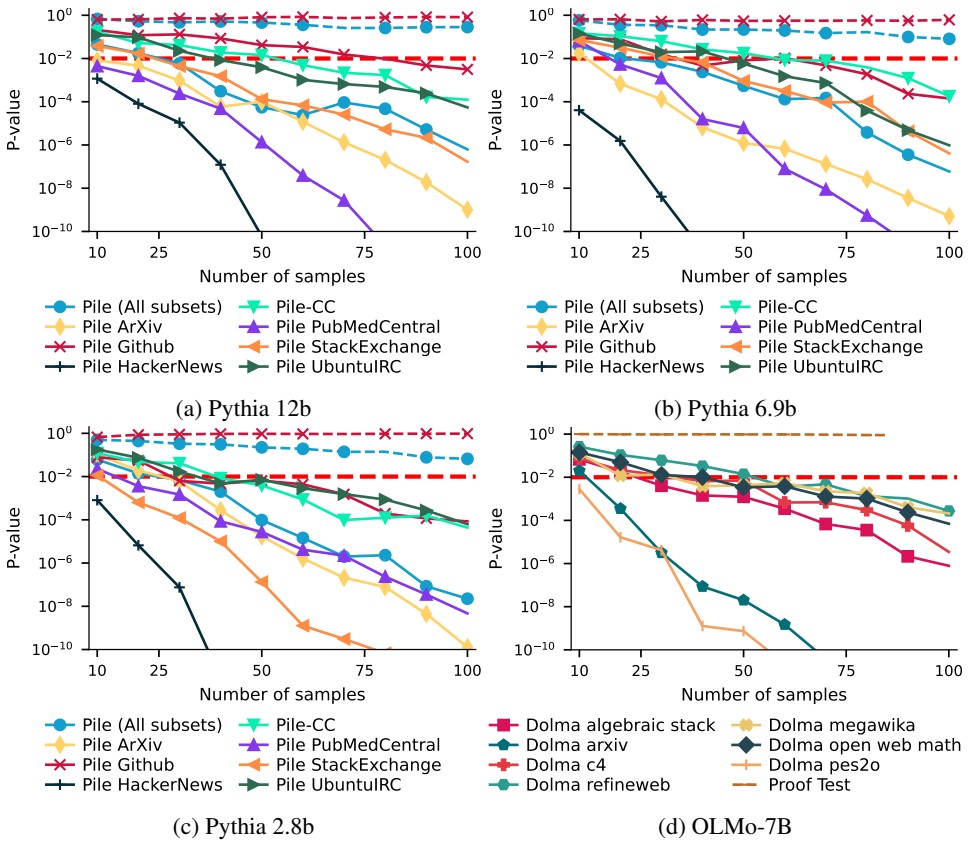

Figure 7: The p-value for different Pythia models and OLMo on subsets of the Pile or Dolma datasets, respectively. We show results for different numbers of samples in the suspect set. For the Pythia models, the solid lines show the training subsets, while the dashed lines are for test subsets (not included in training). The Proof Pile 2 (Test) subset has fewer than 100 NIDs. Hence, their lines are plotted only until the highest number of samples is available. We observe that for training sets, the p-values overall decrease with the number of samples, enabling the detection of the private data in the model's training set. The test set's p-values are constant, suggesting that no false positives are achieved.

## K.3 IMPACT OF MIA STRENGTH

We next investigate how the strength of the underlying membership inference attack affects DI performance with NIDs and, consequently, DP auditing. While our framework treats the MIA as a plug-in component, a stronger MIA signal should intuitively translate into more powerful DI tests. To validate this, we follow the controlled setup: we finetune Pythia-1b on 100 NIDs from the GitHub test set and run DI with four MIA feature configurations. Specifically, we use (1) the original MIA feature set, (2) the original features augmented with CAMIA (Chang et al., 2024), (3) the original features augmented with SURP (Zhang & Wu, 2024), and (4) the combination of original features, CAMIA, and SURP. Table 18 reports the resulting p-values. We observe that incorporating stronger MIAs, particularly CAMIA, substantially improves DI effectiveness, as indicated by lower p-values, and that the trend is consistent: the richer the MIA feature set, the stronger the DI signal.

Table 18: **MIA Strength.** P-values for DI when using different combinations of MIA feature sets, illustrating how stronger MIAs improve the DI signal.

| MIA Signal | P-Value |
|---|---|
| Original Features + CAMIA | $< 1.00 \times 10^{-300}$ |
| Original Features + CAMIA + SURP | $< 1.00 \times 10^{-300}$ |
| Original Features + SURP | $4.17 \times 10^{-211}$ |
| Original Features | $3.31 \times 10^{-156}$ |

### K.4 COMPARING DIFFERENT TYPES OF NIDs

We also study how the structure of an identifier affects DI risk in a controlled experiment (see Table 5 for the exact formats). For each NID format (Java serialization strings, SHA-512, SHA-256, SHA-1, MD5, and Ethereum addresses), we generate a set of identifiers that follow the corresponding pattern, finetune Pythia-1b on 100 instances of that type, and then run DI. Table 19 reports the resulting member p-values. Longer and more structurally complex identifiers, such as SHA-512 hashes, tend to yield stronger DI signals, whereas shorter formats such as MD5 hashes produce weaker but still highly significant results. Beyond length, the character composition also matters: Java serialization strings, which only contains digits, produce a stronger DI signal than SHA-512 despite being shorter. Overall, these results indicate that our framework is robust across a range of realistic NID structures, with DI performance improving as identifiers become more informative and distinctive.

Table 19: **NID Structures.** P-values for DI for different NID formats, showing how identifier length and structure influence the strength of the leakage signal.

| NID Structure | P-Value |
|---|---|
| Java Serialization | $4.17 \times 10^{-211}$ |
| SHA512 | $1.67 \times 10^{-175}$ |
| SHA1 / Ethereum Address | $1.95 \times 10^{-88}$ |
| SHA256 | $8.89 \times 10^{-44}$ |
| MD5 | $7.00 \times 10^{-23}$ |

### K.5 NUMBER OF NIDs

The number of NIDs significantly affects the statistical power of the DI test. To study this effect, we finetune Pythia-1b on 100 NIDs from the GitHub test set, and then run DI using only subsets of size $k \in \{50, 60, 70, 80, 90, 100\}$ of these NIDs. Table 20 shows the resulting member p-values. As expected, the p-values decrease monotonically as the number of NIDs increases, illustrating the sample-complexity benefit of having more identifiers available in the suspect dataset.

Table 20: **Number of NIDs.** P-values for DI as a function of the number of NIDs used, demonstrating the sample-complexity benefit of having more identifiers.

| Number of NIDs | P-Value |
|---|---|
| 50 | $1.01 \times 10^{-66}$ |
| 60 | $7.92 \times 10^{-84}$ |
| 70 | $2.07 \times 10^{-101}$ |
| 80 | $2.23 \times 10^{-119}$ |
| 90 | $1.16 \times 10^{-137}$ |
| 100 | $3.31 \times 10^{-156}$ |

### K.6 TASK-SPECIFIC NIDs

In this subsection, we detail our case study on constructing task-specific NIDs for GSM8K, which consists of grade-school math word problems that require multi-step numerical reasoning. For each selected GSM8K problem, we use GPT-5.1 to rewrite the problem as a numeric template by replacing every concrete number in the statement and solution with a variable; for example, the original solution

fragment "Natalia sold 48/2 = «48/2=24»24 clips in May. Natalia sold 48+24 = «48+24=72»72 clips altogether in April and May. #### 72" is rewritten as the template "Natalia sold $n/2$ clips in May. Natalia sold $n + n/2 = 3n/2$ clips altogether in April and May.", where $n$ stands for the original 48 and all derived quantities become functions of $n$. We then sample new values for these variables, update the problem text and solution accordingly, and have GPT-5.1 verify that each instantiated problem–solution pair is logically correct and self-consistent. In our DI setting, we treat one instance per template as the task-specific NID in the suspect set and the remaining instantiated variants as the corresponding GIDs drawn from the same task-specific distribution. To validate this approach, we finetune Pythia-1b on 100 such GSM8K-derived NIDs and run DI on the resulting suspect sets; as shown in Table 4 in the main paper, these task-specific NIDs enable statistically significant DI on GSM8K. Moreover, the p-values decrease as the number of NIDs increases, reflecting the expected strengthening of the DI signal with additional identifiers.

### K.7 COMPARISON WITH EXISTING DI METHODS

To compare the effectiveness of our method, we not only compare the effectiveness of the individual canaries, but also the performance of the DI methods. For each DI method, we finetune Pythia-1b with the corresponding canaries and apply the corresponding statistical test. Following Maini et al. (2024), we use IID samples and their corresponding statistical test. For Zhang et al. (2024a), we use the Hex and Alphabetic random strings of length 32 and apply our statistical test, as their method lacks one. For Zhao et al. (2025), we still use the entire subset during the generation phase. While this gives an unfair advantage to the method by Zhao et al. (2025), it is necessary to prevent an even larger distribution shift in the resulting generated held-out set. Additionally, the reported time includes both generation and calibration. The generation time is measured in the pre-training setting, on four A100 GPUs, whereas all other experiments use a single A100 GPU.

In Table 21, we report the results from the GitHub subset. We observe that for the member subsets, our method shows strong performance, with lower p-values than Maini et al. (2024) and Zhao et al. (2025). For the non-member subsets, the p-values for all methods are close to 1.0. Notably, the execution time of our approach (21.52 minutes) is close to that of Maini et al. (2024) and the implementation of the approach proposed by Zhang et al. (2024a), yet substantially more efficient than Zhao et al. (2025).

Additionally, in Table 22, we conduct a further evaluation using the whole Pile dataset. In this setting, we are unable to include Zhao et al. (2025), as the method relies on low distributional variability to function effectively. The results show a similar trend to that in the GitHub subset.

Table 21: **DI Comparison (GitHub Subset).** Comparison of DI methods including members/non-members p-values and execution time (in minutes) on the GitHub subset.

| DI Method | P-Value Members | P-Value Non-Members | Time |
|---|---|---|---|
| LLM DI (Maini et al. (2024)) | $9.79 \times 10^{-122}$ | $1.05 \times 10^{-2}$ | 20.43 |
| Unlock DI (Zhao et al. (2025)) | $5.00 \times 10^{-5}$ | $1 \times 10^{0}$ | 2122.37 |
| Zhang et al. (2024a) (Hex) | $7.00 \times 10^{-23}$ | $6.70 \times 10^{-2}$ | 21.18 |
| Zhang et al. (2024a) (Alphabetic) | $< 1.00 \times 10^{-300}$ | $5.42 \times 10^{-1}$ | 20.83 |
| NID DI (Ours) | $3.31 \times 10^{-156}$ | $9.83 \times 10^{-1}$ | 21.52 |

Table 22: **DI Comparison (All Subsets).** Comparison of DI methods including members/non-members p-values and execution time (in minutes) with samples from the Pile.

| DI Method | P-Value Members | P-Value Non-Members | Time |
|---|---|---|---|
| LLM DI (Maini et al. (2024)) | $8.48 \times 10^{-46}$ | $5.85 \times 10^{-1}$ | 20.73 |
| Zhang et al. (2024a) (Hex) | $7.00 \times 10^{-23}$ | $6.70 \times 10^{-2}$ | 21.18 |
| Zhang et al. (2024a) (Alphabetic) | $< 1.00 \times 10^{-300}$ | $5.42 \times 10^{-1}$ | 20.83 |
| NID DI (Ours) | $4.17 \times 10^{-211}$ | $3.76 \times 10^{-1}$ | 20.67 |

## L    DIRECT COMPARISON WITH ZHAO ET AL. (2025)

To further validate our DI method, we compare against Zhao et al. (2025) in the pretrained settings. For fairness, we replicate their experimental setup, including the number of samples reported in Table A2 of (Zhao et al., 2025). We evaluate three subsets of the Pile dataset using the Pythia-6.9b model to ensure coverage across various settings. As shown in Table 23, our method achieves substantially better performance and efficiency, being more effective and orders of magnitude faster than the method of Zhao et al. (2025).

Table 23: **DI Comparison.** Comparison of p-values and end-to-end execution time (in minutes) per subset on Pythia-6.9b.

| Subset | P-Value (Zhao et al., 2025) | P-Value (Ours) | Time (Zhao et al., 2025) | Time (Ours) |
|--------|------|------|------|------|
| Pile-CC | $5.64 \times 10^{-3}$ | $2.18 \times 10^{-34}$ | 1395.87 | 46.17 |
| GitHub | $8.50 \times 10^{-3}$ | $3.65 \times 10^{-14}$ | 2106.97 | 34.41 |
| Ubuntu | $4.23 \times 10^{-2}$ | $3.01 \times 10^{-14}$ | 805.22 | 21.33 |

## M    LIMITATIONS

Our method relies on datasets that contain NIDs. While we have demonstrated that they are widespread, it is possible that not all types of NIDs have been identified; future work may uncover more, which would only enhance our results by increasing the number of real samples.

## N    LLM USAGE

We used LLMs solely to polish author-written text (grammar, clarity, concision). All suggestions were reviewed by the authors, who take full responsibility.

