# OpenReview forum: "Natural Identifiers for Privacy and Data Audits in Large Language Models"
_ICLR.cc/2026/Conference — ICLR 2026 Poster_

### Official Review · Reviewer_aZSB · 2025-10-28

**Soundness:** 3
**Presentation:** 3
**Contribution:** 3
**Rating:** 8
**Confidence:** 5

**Summary:**

This paper introduces Natural Identifiers (NIDs) as a novel mechanism to enable post-hoc privacy and data auditing in large language models (LLMs) without retraining or requiring held-out datasets. The authors define NIDs as structured random strings (e.g., cryptographic hashes, shortened URLs, Ethereum addresses) naturally present in public datasets such as GitHub and StackExchange.
The paper demonstrates that these identifiers can act as natural canaries allowing differential privacy (DP) audits and dataset inference (DI) to be performed post training. It adapts the one run DP auditing framework (Steinke et al., 2023) to work with NIDs and introduces a ranking-based inference procedure that computes tighter DP lower bounds. Extensive experiments on Pythia and OLMo models show that NIDs yield reliable post-hoc DP auditing and DI with no retraining and zero false positives, advancing practical LLM privacy auditing.

**Strengths:**

1. Uses real-world random structures (NIDs) as reusable canaries a clever, previously unexplored idea.
2. Adapts Steinke et al. (2023) into a ranking based auditing theorem (Theorem 1) with rigorous bounds.
3.  Evaluations across models (Pythia, OLMo) and datasets (Pile, Dolma) confirm the scalability and reliability of the method.
4. Eliminates retraining and manual canary insertion, reducing the computational barrier for auditing large LLMs.
5. Directly addresses real world auditing needs in LLM regulation and compliance contexts.

**Weaknesses:**

1. The method assumes datasets contain sufficient NIDs; domains lacking structured identifiers (e.g., medical text) may not benefit.
2.  While appendices are comprehensive, code availability is not stated; releasing tooling for NID extraction would improve accessibility.
4.  Results for smaller ε or limited NIDs could use confidence intervals across more random seeds.
5. A short discussion of potential misuse (e.g., adversarial data reconstruction) would strengthen the ethics section.

**Questions:**

1. How sensitive is DI performance to the number and diversity of NIDs in the suspect dataset?
2. How does the ranking-based ε-bound compare empirically with the two-alternative case when the same number of samples is used?

---

> ### Author Response · Authors · 2025-11-21
>
> We thank the Reviewer for the positive and constructive feedback. We appreciate that our method of using real-world random structures (NIDs) as reusable canaries is considered clever, scalable, rigorous, and reliable. We are glad that our comprehensive experiments clearly demonstrate the practicality of our approach.
>
> We address the concerns in detail one-by-one below:
>
> >**The method assumes datasets contain sufficient NIDs; domains lacking structured identifiers (e.g., medical text) may not benefit.**
>
> We observe that NIDs are not confined to code-oriented or web-centric data. The biomedical corpora, such as the PubMed Central subset of The Pile, contain NIDs. Our work shows that in realistic pretraining corpora, there is typically enough NID signal for effective post-hoc auditing and dataset inference, and extending coverage in lower-NID domains with additional task and domain-specific identifiers (for example, certain grant or trial IDs in scientific text) is a natural direction for future work.
>
> To validate this idea, we design task-specific NIDs for the GSM8K dataset, which consists of grade-school math word problems, and show that they enable effective DI. In the example below (for one of the samples from the GSM8K dataset), we can treat the number 48 and all derived quantities (such as 24 and 72) as a variable and substitute it with different numbers in a consistent way.
>
> Original version (suspect set, **NIDs**):
>
> **Question**
>
> Natalia sold clips to 48 of her friends in April, and then she sold half as many clips in May. How many clips did Natalia sell altogether in April and May?
>
> **Answer**
>
> Natalia sold 48/2 = <<48/2=24>>24 clips in May.
>
> Natalia sold 48+24 = <<48+24=72>>72 clips altogether in April and May.
>
> \#\#\#\# 72
>
> Generated version (held out set, **GIDs**):
>
> **Question**
>
> Natalia sold clips to 46 of her friends in April, and then she sold half as many clips in May. How many clips did Natalia sell altogether in April and May?
>
> **Answer**
>
> Natalia sold 46/2 = <<46/2=23>>23 clips in May.
>
> Natalia sold 46+23 = <<46+23=69>>69 clips altogether in April and May.
>
> \#\#\#\# 69
>
>
> To validate this approach, we finetune Pythia-1b on 100 NIDs based on GSM8K.
> To generate the GIDs, we prompt GPT-5.1 to first solve the task and then generate a general template with a parametric and imputable solution. Then, we generate 127 new identifiers and verify that the generated identifiers are correct by evaluating them via zero-shot prompting using GPT-5.1. The table below shows that this new type of task-specific NID can be effectively applied to DI.
>
> |   Number of NIDs|   P-Value |
> |-----:|---:|
> |50 |          8.43e-04 |
> |60 |          9.56e-05 |
> |70 |          3.35e-04 |
> |80 |          1.63e-05 |
> |90 |          2.12e-06 |
> |           100 |          1.60e-06 |
>
> We added into the paper the table above (Table 4) with the corresponding results, a short paragraph in Section 5 discussing them, and Appendix K.6, which describes the experimental setup.
>
> >**While appendices are comprehensive, code availability is not stated; releasing tooling for NID extraction would improve accessibility.**
>
> The code is available in the supplementary material, along with a detailed explanation of how to use it. Moreover, the dockerfile allows an easy setup of the Python environment.
> To run the NID extraction, the first step, the regex extraction, is done in `run_regex.py`. This script can be easily extended by adding new datasets (by changing the lines after line 111) and by adding new types of NIDs by modifying the dict at line 30. After that, a second filtering step is done in `blind_baseline_step_1.py` and `blind_baseline_step_2.py`. After executing these scripts, the resulting output is `filtered_step_2`.
>
> >**Results for smaller $\varepsilon$ or limited NIDs could use confidence intervals across more random seeds.**
>
> Both the auditing of differential privacy and dataset inference produce statistically sound p-values and confidence intervals. For completeness, we report the estimated $\varepsilon$ for Pythia-1b using 4 seeds. We added these results to Table 7 in the new Appendix H.2 and present the results below as well:
>
> |Cardinality| Real $\varepsilon$|Estimated $\varepsilon$|
> |:---:|:--:|---|
> |2|5|0.086 $\pm$ 0.021|
> |2|10|0.106 $\pm$ 0.020|
> |2|100|0.245 $\pm$ 0.023|
> |2|$\infty$|0.979 $\pm$ 0.028|
> |8|5|0.720 $\pm$ 0.144|
> |8|10|0.789 $\pm$ 0.144|
> |8|100|1.094 $\pm$ 0.129|
> |8|$\infty$|2.329 $\pm$ 0.058|
> |32|5|1.761 $\pm$ 0.226|
> |32|10|1.830 $\pm$ 0.231|
> |32|100|2.178 $\pm$ 0.218|
> |32|$\infty$|3.775 $\pm$ 0.067|
>
> Across all settings, the estimated $\varepsilon$ values are highly consistent over 4 random seeds, with relatively small standard deviations (at most $\approx 0.23$) compared to the mean estimates.

---

> > ### Author Response · Authors · 2025-11-21
> >
> > >**A short discussion of potential misuse (e.g., adversarial data reconstruction) would strengthen the ethics section.**
> >
> > We expanded the ethics section with a discussion of dual use. In particular, we noted that while our goal is to enable oversight and privacy auditing, the same techniques could in principle be misused to better locate training artifacts or aid reconstruction attacks.
> >
> > >**How sensitive is DI performance to the number and diversity of NIDs in the suspect dataset?**
> >
> > The more samples, the higher the power of the statistical test. Regarding the diversity of the NIDs, we ran an additional comprehensive microanalysis on the existing types of NIDs to answer the question by finetuning a Pythia-1b on each synthetic NID that follows the specific NID pattern by adding 100 canaries to precisely measure their impact.
> >
> > Below, the first table reports for each NID format. Overall, we observe that each format has quite a diverse impact on the privacy leakage, from Java serialization, which leaks the most, to MD5, which is the least vulnerable to DI.
> >
> >
> > | NID structure          |   P-Value|
> > |:--|---:|
> > | Java Serialization     |         4.17e-211 |
> > | SHA512     |         1.67e-175 |
> > | SHA1 / Ethereum Wallet |         1.95e-88  |
> > | SHA256     |         8.89e-44  |
> > | MD5        |         7.00e-23  |
> >
> > In addition, using the same setup, we finetuned Pythia-1b on the GitHub test-set NIDs and evaluated the effect of varying the number of samples. The results are shown below. We observe that as the number of NIDs increases, the p-value decreases.
> >
> > |   Number of NIDs |   P-Value|
> > |--:|:--:|
> > |50 |         1.01e-66  |
> > |60 |         7.92e-84  |
> > |70 |         2.07e-101 |
> > |80 |         2.23e-119 |
> > |90 |         1.16e-137 |
> > |           100 |         3.31e-156 |
> >
> >
> > We added these results to the new Appendix K.4 and K.5, respectively. Moreover, we discuss and analyze them in a new paragraph in Section 5.
> >
> >
> > >**How does the ranking-based $\varepsilon$-bound compare empirically with the two-alternative case when the same number of samples is used?**
> >
> > For the randomized response (Figure 2), we show that higher cardinality (i.e., more generated GIDs than two alternatives) is especially beneficial at larger privacy budgets ($\varepsilon \ge 8$), which is the typical regime in LLM training with DP. Thus, our framework scales naturally with the number of GIDs. Concretely, using 100 NIDs and with the theoretical upper bound $\varepsilon=10$, the estimated $\varepsilon$ by the two-alternatives is only roughly between 0.3 and 0.7. On the other hand, our 32 GIDs give a much tighter estimate of roughly between 4.5 and 6.
> >
> > Similarly, the empirical analysis in Figure 3 demonstrates that our method with the ranking-based approach outperforms the baseline with two alternatives across multiple cardinality parameters ($c \in \\{8, 32\\}$) in fixed-length dataset settings. Finally, we theoretically analyze the ranking-based approach in Appendix F, by applying the Bernstein inequality and analyzing the asymptotic behaviour of the concentration margin.
> >
> > Overall, increasing cardinality in our ranking-based approach makes the privacy auditing more precise and tighter, particularly in less restrictive privacy settings (higher $\varepsilon$ values).
> >
> >
> > We hope that these answers addressed the reviewer’s concerns and would be happy to provide further clarifications if needed.

---

### Official Review · Reviewer_uy2q · 2025-10-28

**Soundness:** 3
**Presentation:** 3
**Contribution:** 3
**Rating:** 8
**Confidence:** 3

**Summary:**

The paper introduces natural identifiers (NIDs), ie random-looking strings common on the web (such as hashes, short URLs, eth addresses) as a scalable basis for post-hoc privacy auditing and dataset inference in LLMs. As NIDs have known generation rules, the authors mint matched generated identifiers (GIDs) to serve as iid controls, replacing bespoke canaries. This enables a single-run DP audit (picked up from the literature) that ranks NIDs vs. GIDs to yield tighter lower bounds with fewer samples, and a dataset-inference test that flags whether a suspect corpus was in training using only that corpus plus GIDs. Experiments on Pythia/OLMo and standard datasets show accurate hits on true training subsets and no false positives on held-out data. This makes NIDs a practical, general auditing tool.

**Strengths:**

The paper repurposes naturally occurring structured random strings as audit beacons and then samples from their known generators to create IID controls; this is a simple but fresh reframing that removes the retraining crutch many prior audits depend on, and I found it genuinely interesting and surprising; the ranking-based adaptation of one-run DP auditing further broadens that idea beyond top-1 decisions and tightens the link to theory.

The work further formalizes NIDs/GIDs and their sampling assumptions, connects the audit to an epsilon-DP theorem, and then evaluates on models with known training data (Pythia, OLMo) where ground truth is verifiable, reporting consistent DI behavior with no false positives on held-out subsets and showing how cardinality affects statistical power.

In terms of clarity, the exposition is well structured: the paper motivates why IID held-out data is the bottleneck, illustrates NIDs with concrete examples (e.g hashes, shortened URLs, crypto addresses), and uses clean figures and notation to map the end-to-end pipeline from NID extraction to GID generation to ranked auditing.

Overall I believe the paper's significance is credible, because NIDs are abundant across standard pretraining corpora, making the method deployable in practice and potentially valuable for post-hoc audits or discovery in contentious settings.

**Weaknesses:**

In my view, the paper's main issue is that it depends on artifacts that are trivially removable at ingest: hashes, wallet addresses, shortened URLs, and similar strings are easy for a data curator to strip, normalize, or mask with a one-line regex during scraping or deduplication. Given that the method's leverage comes from these rigid formats and abundance, a mild shift in preprocessing policy could sharply reduce coverage, making the technique short-lived in practice. The authors show that many current corpora contain thousands of such identifiers (especially code-heavy subsets like GitHub/StackExchange and broad crawls like RefinedWeb) but those same tables also reveal how uneven the distribution is across domains, foreshadowing fragility under even modest curation pressure. A responsible next step would be to measure how counts and audit power change under realistic filtering pipelines (newline/HTML stripping, de-tokenization, heuristic hash removal, URL canonicalization), and to report DI and epsilon lower bounds before/after such defenses on the same models and suspect sets.

**Questions:**

1. How robust is the approach to trivial preprocessing that filters or normalizes NIDs? Today's training pipelines often apply regex-based scrubbing and URL canonicalization; because your extraction relies on highly rigid patterns (e.g., [a-fA-F0-9]{32,128}, 0x[a-fA-F0-9]{40}), a curator could remove or mask these in one pass. Please quantify the drop in NID counts and DI/DP-audit power after simulating common cleaning steps. A table like the distribution-of-NIDs summary, but "before vs. after" realistic filters, would show whether the method degrades to impracticality with minimal countermeasures.

2. Are GIDs sampled from the exact same effective distribution as the observed NIDs, not just the generator's support? Many NIDs in the wild reflect non-uniformities (e.g., address casing biases, popular shorteners, repository-specific hash exposure).

3. Is it possible to analyze failure modes where small generator inaccuracies leak membership signal? For example, Ethereum checksums, preferred hash casing patterns, or URL-shortener path semantics might let the model distinguish NIDs from your GIDs.

---

> ### Author Response · Authors · 2025-11-21
>
> We thank the Reviewer for the positive and detailed feedback. We appreciate that the Reviewer finds our auditing tool scalable, practical, and general, while writing is clear and the problem well-motivated.
>
> In summary, even highly filtered and curated corpora such as Dolma still contain a substantial number of NIDs. Furthermore, for each specific subdomain, it is possible to design NIDs that are specifically tailored to the given dataset.
>
> We provide detailed answers in-line below:
>
> > **In my view, the paper's main issue is that it depends on artifacts that are trivially removable at ingest: hashes, wallet addresses, shortened URLs, and similar strings are easy for a data curator to strip, normalize, or mask with a one-line regex during scraping or deduplication. [...]**
>
> Dolma’s web subset is a concrete example of a highly curated corpus that already applies the kind of comprehensive preprocessing the reviewer describes. Starting from Common Crawl, it undergoes a CCNet-style pipeline, multiple quality filters, regex- and classifier-based PII and safety filters, and multi-stage deduplication. Despite this extensive cleaning, we still observe tens of thousands of distinct NID types and millions of occurrences, and all our DI and DP-audit results are obtained in this filtered regime. Completely removing NIDs would require a level of effort well beyond standard hygiene such as URL canonicalization or basic regex filtering. A model developer would need to identify and remove every NID type (even newly emerging ones) from a multi-trillion-token corpus. By contrast, the auditor only needs roughly 100 surviving NIDs and can update their detectors post-hoc, making our approach far more robust against even adversarial curation.
> Additionally, when NIDs are scarce, it is often possible to design task-specific approaches, as in the case of the GSM8K dataset. To validate this idea, we design task-specific NIDs for the GSM8K dataset, which consists of grade-school math word problems, and show that they enable effective DI. In the example below (for one of the samples from the GSM8K dataset), we can treat the number 48 and all derived quantities (such as 24 and 72) as a variable and substitute it with different numbers in a consistent way.
>
> Original version (suspect set, **NIDs**):
>
> **Question**
>
> Natalia sold clips to 48 of her friends in April, and then she sold half as many clips in May. How many clips did Natalia sell altogether in April and May?
>
> **Answer**
>
> Natalia sold 48/2 = <<48/2=24>>24 clips in May.
>
> Natalia sold 48+24 = <<48+24=72>>72 clips altogether in April and May.
>
> \#\#\#\# 72
>
> Generated version (held out set, **GIDs**):
>
> **Question**
>
> Natalia sold clips to 46 of her friends in April, and then she sold half as many clips in May. How many clips did Natalia sell altogether in April and May?
>
> **Answer**
>
> Natalia sold 46/2 = <<46/2=23>>23 clips in May.
>
> Natalia sold 46+23 = <<46+23=69>>69 clips altogether in April and May.
>
> \#\#\#\# 69
>
>
> To validate this approach, we finetune Pythia-1b on 100 NIDs based on GSM8K.
> To generate the GIDs, we prompt GPT-5.1 to first solve the task and then generate a general template with a parametric and imputable solution. Then, we generate 127 new identifiers and verify that the generated identifiers are correct by evaluating them via zero-shot prompting using GPT-5.1. The table below shows that this new type of task-specific NID can be effectively applied to DI, with all reported p-values indicating strong statistical significance for every tested number of NIDs.
>
> |   Number of NIDs|   P-Value |
> |-----:|---:|
> |50 |          8.43e-04 |
> |60 |          9.56e-05 |
> |70 |          3.35e-04 |
> |80 |          1.63e-05 |
> |90 |          2.12e-06 |
> |           100 |          1.60e-06 |
>
> We added into the paper the table above (Table 4) with the corresponding results, a short paragraph in Section 5 discussing them, and Appendix K.6, which describes the experimental setup.

---

> > ### Author Response · Authors · 2025-11-21
> >
> > > **How robust is the approach to trivial preprocessing that filters or normalizes NIDs? Today's training pipelines often apply regex-based scrubbing and URL canonicalization; because your extraction relies on highly rigid patterns (e.g., [a-fA-F0-9]{32,128}, 0x[a-fA-F0-9]{40}), a curator could remove or mask these in one pass. Please quantify the drop in NID counts and DI/DP-audit power after simulating common cleaning steps. A table like the distribution-of-NIDs summary, but "before vs. after" realistic filters, would show whether the method degrades to impracticality with minimal countermeasures.**
> >
> > Dolma already represents a strongly preprocessed corpus with regex-based scrubbing, deduplication, and canonicalization, so our evaluation effectively reflects an “after-filtering” setting. While we did not simulate additional synthetic filters, we clarified this point and discussed the adversarial curator scenario in Section 3.1. We report the number of identified NIDs in the highly curated Dolma in Table 6 (Appendix D).  In practice, maintaining exhaustive filters to suppress every NID type is far more demanding than detecting a small surviving subset, which makes our auditing resilient under realistic modern data-curation pipelines.
> >
> > > **Are GIDs sampled from the exact same effective distribution as the observed NIDs, not just the generator's support? Many NIDs in the wild reflect non-uniformities (e.g., address casing biases, popular shorteners, repository-specific hash exposure).**
> >
> > We carefully sample GIDs to closely match the empirical distribution of the observed NIDs. Specifically, we preserve casing patterns, remove duplicates, and use a blind baseline (a model-agnostic classifier trained only on textual features to ensure real and generated identifiers are indistinguishable without model access) to filter out biased or invalid samples. The full sampling and filtering procedure is detailed in Appendix C.
> >
> >
> > > **Is it possible to analyze failure modes where small generator inaccuracies leak membership signal? For example, Ethereum checksums, preferred hash casing patterns, or URL-shortener path semantics might let the model distinguish NIDs from your GIDs.**
> >
> > A wrong and careless implementation of the generator would produce a distribution shift between NIDs and GIDs. This would produce false positives for DI and an overestimation of the $\varepsilon$ for DP auditing.
> >
> > To quantify these failure modes, we finetuned a Pythia-1b on a random hexadecimal sequence of 32 characters to resemble an MD5. We consider different ways to corrupt the GID generator. We consider the following types of misimplementation of the generator: wrong casing (the NIDs are lowercase, while the GIDs are upper case) and off by X (the GIDs are X more random characters than the NID).
> >
> > | Canary Type     |   P-Value Members |   P-Value Non-Members |
> > |:-|---:|---:|
> > | Wrong Case      |         <1.00e-300  | 1.16e-54 |
> > | Length Off By 2 |         3.64e-99  | 5.39e-02 |
> > | Length Off By 1 |         <1.00e-300 | 3.40e-01 |
> > | Default         |         3.31e-156 | 9.83e-01 |
> >
> > We observe that in this case, some types of misimplementations such as wrong casing can cause false positives. However, the MIAs signal might be a bit more robust to some other types of errors, such as adding a small number of tokens. This robustness might be caused by strong MIA signals such as Min-K% and Min-K%++, that only consider a subset of the tokens.
> > We added a discussion of these new results in a new paragraph in Section 5, and added them to the new Appendix K.2.
> >
> >
> > We hope that these answers addressed the reviewer’s concerns and would be happy to provide further clarifications if needed.

---

> > > ### Comment · Reviewer_uy2q · 2025-11-27
> > >
> > > Thank you for your answers! Apologies for missing the information that was already included in the Appendix. I confirm my initial score to accept the paper to the conference.

---

> > > > ### Author Response · Authors · 2025-11-28
> > > >
> > > > We thank the Reviewer for the time they dedicated to evaluating our submission, and we remain available to address any further questions that may arise.

---

### Official Review · Reviewer_TZKy · 2025-11-10

**Soundness:** 3
**Presentation:** 3
**Contribution:** 3
**Rating:** 6
**Confidence:** 3

**Summary:**

This work introduces using natural identifiers as a novel way to audit data privacy in large language models. Natural identifiers are structured random strings such as the output of hash algorithms, shortened URLs, and crypto addresses, which are (i) naturally included in datasets such as Pile and Dolma and (ii) easy to generate held-out data from the same distribution. The authors demonstrates the effectiveness of this idea thru post-hoc DP audinting without training and practical dataset inference.

**Strengths:**

1. The idea is clear and well-motivated, smartly tackling the key problem of how to find held-out data from the same distribution.
2. The authors provide extensive empirical experiments, including ε estimation and dataset inference, to demonstrate the effectiveness of using natural identifiers in real-world scenarios.

**Weaknesses:**

1. The limitation of natural identifiers is obvious: Will this kind of data continually exist in pretraining data? In other words, if one model developer does not want their models to be audited, it is fairly easy for them to remove the natural identifier without sacrificing the model quality.
2. For dataset inference, it requires the audited dataset to contain natural identifiers, which could be hard constraint. For example, if someone want to test whether GSM8K is contained in training, it seems hard to use natural identifiers.

**Questions:**

See Weaknesses

---

> ### Author Response · Authors · 2025-11-21
>
> We thank the Reviewer for the positive feedback. We appreciate that the Reviewer finds our idea clear and well-motivated, and that our method is described as smart.
>
> In summary, even highly filtered and curated corpora such as Dolma still contain a substantial number of NIDs. Furthermore, for each specific subdomain, it is possible to design NIDs that are specifically tailored to the given dataset.
>
> We provide detailed answers in-line below:
>
> >**The limitation of natural identifiers is obvious: Will this kind of data continually exist in pretraining data? In other words, if one model developer does not want their models to be audited, it is fairly easy for them to remove the natural identifier without sacrificing the model quality.**
>
> Even highly filtered and curated corpora such as Dolma still contain a substantial number of NIDs (as we show in Table 6 Appendix D, Dolma alone has 23571 distinct NID types), and new formats keep appearing, which makes reliably stripping all of them during preprocessing practically very difficult. This creates a strong asymmetry in favor of the auditor: the model developer would need to identify and remove every NID type from a multi-trillion-token corpus, while the auditor only needs to find on the order of 100 NIDs in the suspect set and can continually update their detectors as new NID formats emerge after the model has been released, while the model developer can no longer do that. Our experiments, therefore, focus on the realistic setting of web-trained LLMs where NIDs are pervasive in practice and provide a stable signal for auditing. Additionally, when NIDs are scarce, it is often possible to design task and domain-specific approaches, as in the case of the GSM8K dataset (see the next answer for the detailed description).
>
> >**For dataset inference, it requires the audited dataset to contain natural identifiers, which could be hard constraint. For example, if someone wants to test whether GSM8K is contained in training, it seems hard to use natural identifiers.**
>
> It is possible to design NIDs that are specifically tailored to the given dataset. To showcase this feasibility, we design dataset-specific NIDs for the GSM8K dataset. For instance, in the example below (for one of the samples from the GSM8K dataset), we can treat the number 48 and all derived quantities (such as 24 and 72) as a variable and substitute it with different numbers in a consistent way.
>
> Original version (suspect set, **NIDs**):
>
> **Question**
>
> Natalia sold clips to 48 of her friends in April, and then she sold half as many clips in May. How many clips did Natalia sell altogether in April and May?
>
> **Answer**
>
> Natalia sold 48/2 = <<48/2=24>>24 clips in May.
>
> Natalia sold 48+24 = <<48+24=72>>72 clips altogether in April and May.
>
> \#\#\#\# 72
>
> Generated version (held out set, **GIDs**):
>
> **Question**
>
> Natalia sold clips to 46 of her friends in April, and then she sold half as many clips in May. How many clips did Natalia sell altogether in April and May?
>
> **Answer**
>
> Natalia sold 46/2 = <<46/2=23>>23 clips in May.
>
> Natalia sold 46+23 = <<46+23=69>>69 clips altogether in April and May.
>
> \#\#\#\# 69
>
>
> To validate this approach, we finetune Pythia-1b on 100 NIDs based on GSM8K.
> To generate the GIDs, we prompt GPT-5.1 to first solve the task and then generate a general template with a parametric and imputable solution. Then, we generate 127 new identifiers and verify that the generated identifiers are correct by evaluating them via zero-shot prompting using GPT-5.1. The table below shows that this new type of task-specific NID can be effectively applied to DI, with all reported p-values indicating strong statistical significance for every tested number of NIDs and showcasing the general applicability of our framework.
>
> |   Number of NIDs|   P-Value |
> |-:|-:|
> |50 |8.43e-04 |
> |60 |9.56e-05 |
> |70 |3.35e-04 |
> |80 |1.63e-05 |
> |90 |2.12e-06 |
> |100 |1.60e-06 |
>
> We added into the paper the table above (Table 4) with the corresponding results, a short paragraph in Section 5 discussing them, and Appendix K.6, which describes the experimental setup.
>
> If the above answers address the Reviewer’s concerns, we would greatly appreciate if they could consider updating their score.

---

> ### Comment · Reviewer_TZKy · 2025-11-21
>
> Thanks to the authors for their response.
>
> The added discussion addressed my concern, and I've increased my score.

---

> > ### Author Response · Authors · 2025-11-22
> >
> > We are grateful for your thoughtful reassessment and the improved score, and we thank you for the time invested in reviewing our submission. Should any further questions arise, we remain available to answer.

---

### Official Review · Reviewer_6fad · 2025-11-11

**Soundness:** 3
**Presentation:** 3
**Contribution:** 2
**Rating:** 2
**Confidence:** 3

**Summary:**

This paper studies post-hoc privacy audits for evaluating and quantifying LLMs privacy risks.
It targets (i) auditing differential privacy (DP) claims: empirically estimating a lower bound on the DP budgets and (ii) dataset inference (DI): inferring whether an entire data subset was used to train the model.

The key idea is to use Natural Identifiers (NIDs)—structured, random-looking strings (e.g., hashes, shortened URLs) that naturally appear in the training set—and to generate additional strings from the same distribution. NIDs enable the generation of unlimited additional random strings from the same distribution, serve both as canaries for DP auditing and as same-distribution held-out data for DI.

**Strengths:**

- Studies an important problem of LLM privacy audit

- Good articulation of the limits of prior privacy audits

**Weaknesses:**

1. Dedicated injected canaries vs NIDs. It’s unclear how results depend on which NIDs you pick. Prior work uses dedicated canaries injected from out of the training set; yours are random strings within the training set. Please quantify whether in-distribution vs OOD canaries change audit power and bias.

2. Why tighter DP lower bounds / lower sample complexity? You reported "Our privacy auditing with NIDs improves the lower bounds on the privacy parameters of an algorithm compared to the auditing framework by Steinke et al. (2023). It also significantly reduces the sample complexity, i.e., it requires fewer NID canaries."

3. DI performance/gains look modest. Is this because of the inherent limitation of your work in which you infer whether an entire data subset was used to train the model by reasoning only on NIds within the dataset, not the whole dataset?

4. Comparisons missing with Zhang et al., 2024 and Zhao et al., 2025 for DI (both performance and efficiency) to support your claims against these prior works. Also experiments lack the use of more recent MIAs.

5. Scope / external validity. Results rely on Pythia/Pile and OLMo/Dolma. Where do NIDs not exist? Provide a coverage analysis and a failure mode.

6. Over-reach beyond LLMs. You mention DI for Diffusion and Image Autoregressive Models "Beyond LLMs, DI has also been successfully applied to other types of generative models, including Diffusion Models (Dubinski et al., 2025) and ´ Image Autoregressive Models (Kowalczuk et al., 2025), widening the potential of the method." but don’t evaluate them. Either add experiments or tone down claims.

7. Lack significant technical contribution wrt to the closest prior work. As you report "Zhang et al. (2024a) propose to inject random and meaningless canaries into the data and then test how the LLM ranks the selected canary among all alternatives."
Relative to Zhang et al. (injected random strings), your technical contribution is switching to NIDs that are naturally included in LLMs’ training sets. I would note this earlier in the paper, e.g., in the introduction too.

**Questions:**

See above.

---

> ### Author Response · Authors · 2025-11-21
>
> We thank the Reviewer for the detailed analysis of our paper. We appreciate that the Reviewer finds the topic of our paper important and its clear positioning within the current literature on privacy audits.
>
> In summary, we generalize the method of Steinke et al. 2023 to the zero-run (post-hoc) privacy auditing and improve it by using higher cardinalities. Our approach provides the held-out set for DI from the exact same distribution as the suspect set, which is the truly valid and most challenging setting.
>
> We provide detailed answers in-line below:
>
> >**Dedicated injected canaries vs NIDs. It’s unclear how results depend on which NIDs you pick. Prior work uses dedicated canaries injected from out of the training set; yours are random strings within the training set. Please quantify whether in-distribution vs OOD canaries change audit power and bias.**
>
> Our setting explicitly assumes that we *cannot* inject dedicated canaries into the training pipeline, which is a key difference from prior work (Steinke et al. 2023) that controls the data generation process and sampling of the canaries. In our new zero-run (post-hoc) regime, we automatically select NIDs that are naturally already included in the training data using fixed detectors, so our results reflect the aggregate behavior of many NIDs under a realistic auditing workflow.
>
> To further address the reviewer’s comment, we ran additional experiments to compare the strength of our NIDs vs. different types of manually inserted canaries. While we cannot run a controlled comparison by fully training a model from scratch due to the computational cost, following [1], we finetuned Pythia-1b as a proxy, on 100 canaries.
> In particular, we evaluate four canary types: (1) random alphabetic strings of length 32, (2) NIDs extracted from the GitHub subset of the Pile test set, (3) in-distribution text from the Pile test set, and (4) random hexadecimal strings of length 32.
> Although those  injected canaries fall outside our post-hoc threat model, this controlled setting helps contextualize the strength of the NID leakage relative to existing auditing methods.
>
> The results below show the p-values obtained for the different types of canary. Overall, we observe that NIDs remain competitive with other injected canary types. In particular, NIDs are better at capturing privacy leakage than IID canaries, but slightly worse than some types of crafted canaries, such as alphabetic, but better than hexadecimal canaries. Still, our NIDs exhibit the significant conceptual advantage over all injected canaries that they allow for *post-hoc* auditing without having to retrain/fine-tune a model.
>
>
> | Canary Type|P-Value|
> |:-|-:|
> | Alphabetic|<1.00e-300  |
> | NIDs|3.31e-156 |
> | IID|4.55e-100 |
> | Hex|7.00e-23  |
>
>
> We have included these results in Table 16, alongside the relevant discussion in Appendix K.1. We have also referenced them in a new paragraph in Section 5.

---

> > ### Author Response · Authors · 2025-11-21
> >
> > > **Why tighter DP lower bounds / lower sample complexity? You reported "Our privacy auditing with NIDs improves the lower bounds on the privacy parameters of an algorithm compared to the auditing framework by Steinke et al. (2023). It also significantly reduces the sample complexity, i.e., it requires fewer NID canaries."**
> >
> > Steinke et al. (2023) is a special case of our Theorem 1, where the cardinality is 2. The intuition is that with higher cardinality (more GIDs per an NID), correctly guessing becomes a harder task, thus giving stronger evidence of knowledge of the training data.
> >
> > On the empirical side, in Figure 2, we analyze this behaviour in the randomized response setting, where a higher $c$ requires lower sample complexity. For instance, if we ask how many samples are needed to show that $\varepsilon$ is at least 5, we would need roughly 1000 samples when the cardinality is 2 (as in Steinke et al. 2023), while only roughly 30 when the cardinality is 32 (as possible with our approach).
> >
> > Moreover, in Appendix F, we explain it in a more theoretically grounded and precise manner. In particular, if we apply the Bernstein inequality, the concentration margin roughly decays as $\sqrt\frac{1}{mc}$, where $m$ is the number of NIDs and $c$ is the cardinality.
> >
> >
> > > **DI performance/gains look modest. Is this because of the inherent limitation of your work in which you infer whether an entire data subset was used to train the model by reasoning only on NIDs within the dataset, not the whole dataset?**
> >
> > Our DI gains may look modest in absolute terms, but they are obtained in a more constrained and fully post-hoc setting with the *perfectly IID* held-out set. The existing work (Maini et al., 2024; Zhao et al., 2025) typically uses 1000–2000 samples to obtain statistically significant results (p < 0.01), whereas we achieve this with only 100 NIDs, enabled by our ranking-based statistical test.
> >
> > There is no inherent limitation of performing DI only on NIDs. Instead, these NIDs can be considered as strong evidence, similarly to canaries in privacy auditing. Our framework can generate an unbounded number of GIDs (no limitation on the held-out data size) and forms the held-out sets from the exact same distribution as the suspect set, which is a strictly more difficult task than when there is a distribution shift between the suspect and held-out sets (Maini et al., 2024; Zhao et al., 2025). Ultimately, our method correctly identifies all training subsets (true positives) and all non-training subsets (true negatives), as shown in Tables 2 and 3 of the main paper.
> >
> >
> >
> > > **Comparisons missing with Zhang et al., 2024 and Zhao et al., 2025 for DI (both performance and efficiency) to support your claims against these prior works.**
> >
> > Zhang et al., 2024 propose to inject the canaries in the training data before training the model. Therefore, it cannot be applied post-hoc, which severely limits its applicability.
> >
> > Zhao et al. (2025) construct a held-out set that exhibits distribution shifts relative to the suspect set, leading to false positives and necessitating post-hoc calibration. In contrast, our method creates the held-out set from the **exact same distribution** as the suspect set, eliminating the distribution shifts and preventing false positives without requiring additional calibration.
> >
> > Moreover, Zhao et al., 2025’s approach is computationally expensive, requiring extensive training and calibration. Our method is faster and less costly. In particular, the training of the generator in Zhao et al., 2025 requires 6-10 hours on 4 A100 GPUs for each subset. The extraction of the NIDs and the generation of the corresponding GIDs is significantly cheaper, and on the GPU side, it only requires a 1-2 forward pass(es) per sample (less than 10 minutes on 4 A100 GPUs for each subset).

---

> > > ### Author Response · Authors · 2025-11-21
> > >
> > > >**Also experiments lack the use of more recent MIAs.**
> > >
> > > Our work is orthogonal to the performance of the individual MIAs. Designing or leveraging more effective MIAs translates to better DI and tighter DP auditing. To validate this intuition, following the same controlled evaluation setup described above (Appendix K.3), we ran additional experiments and fine-tuned Pythia-1b on 100 NIDs. Then, we run DI using our original MIA features and our original MIA **plus** CAMIA [2] and SURP [3], two recent MIAs for LLMs.
> > >
> > > The table below reports the p-value under different sets of MIA signals for DI. We observe that a stronger MIAs, such as CAMIA, significantly improve DI effectiveness (lower p-value). Moreover, the better the MIA signal, the larger the improvement.
> > >
> > > | MIA Signal|   P-Value|
> > > |:---|---:|
> > > | Original Features  + CAMIA|<1.00e-300  |
> > > | Original Features  + CAMIA + SURP |<1.00e-300  |
> > > | Original Features  + SURP|4.17e-211 |
> > > | Original Features|3.31e-156 |
> > >
> > > We have added the detailed discussion and this table in Appendix K.3 (Table 18).
> > >
> > >
> > >
> > > > **Scope / external validity. Results rely on Pythia/Pile and OLMo/Dolma. Where do NIDs not exist? Provide a coverage analysis and a failure mode.**
> > >
> > > Our empirical coverage analysis in Table 6 in Appendix D  demonstrates that NIDs are abundant in many subsets of Pile and Dolma. Our focus is on realistic web-trained LLMs where NIDs are common. Extending coverage in low-NID domains by discovering additional domain-specific identifiers is an interesting direction.
> > > To explore this direction, we design task-specific NIDs for the GSM8K dataset, which consists of grade-school math word problems, and show that they enable effective DI. In the example below (for one of the samples from the GSM8K dataset), we can treat the number 48 and all derived quantities (such as 24 and 72) as a variable and substitute them with different numbers in a consistent way.
> > >
> > > Original version (suspect set, **NIDs**):
> > >
> > > **Question**
> > >
> > > Natalia sold clips to 48 of her friends in April, and then she sold half as many clips in May. How many clips did Natalia sell altogether in April and May?
> > >
> > > **Answer**
> > >
> > > Natalia sold 48/2 = <<48/2=24>>24 clips in May.
> > >
> > > Natalia sold 48+24 = <<48+24=72>>72 clips altogether in April and May.
> > >
> > > \#\#\#\# 72
> > >
> > > Generated version (held out set, **GIDs**):
> > >
> > > **Question**
> > >
> > > Natalia sold clips to 46 of her friends in April, and then she sold half as many clips in May. How many clips did Natalia sell altogether in April and May?
> > >
> > > **Answer**
> > >
> > > Natalia sold 46/2 = <<46/2=23>>23 clips in May.
> > >
> > > Natalia sold 46+23 = <<46+23=69>>69 clips altogether in April and May.
> > >
> > > \#\#\#\# 69
> > >
> > > To validate this approach, we finetune Pythia-1b on 100 NIDs based on GSM8K.
> > > To generate the GIDs, we prompt GPT-5.1 to first solve the task and then generate a general template with a parametric and imputable solution. Then, we generate 127 new identifiers and verify that the generated identifiers are correct by evaluating them via zero-shot prompting using GPT-5.1. The table below shows that this new type of task-specific NID can be effectively applied to DI, with all reported p-values indicating strong statistical significance for every tested number of NIDs.
> > >
> > > |   Number of NIDs|   P-Value |
> > > |-:|-:|
> > > |50 |8.43e-04 |
> > > |60 |9.56e-05 |
> > > |70 |3.35e-04 |
> > > |80 |1.63e-05 |
> > > |90 |2.12e-06 |
> > > |100 |1.60e-06 |
> > >
> > > We added into the paper the table above (Table 4) with the corresponding results, a short paragraph in Section 5 discussing them, and Appendix K.6, which describes the experimental setup.

---

> > > > ### Author Response · Authors · 2025-11-21
> > > >
> > > > > **Over-reach beyond LLMs. You mention DI for Diffusion and Image Autoregressive Models "Beyond LLMs, DI has also been successfully applied to other types of generative models, including Diffusion Models (Dubinski et al., 2025) and ´ Image Autoregressive Models (Kowalczuk et al., 2025), widening the potential of the method." but don’t evaluate them. Either add experiments or tone down claims.**
> > > >
> > > > We did not intend to claim that our NID-based auditing method has been applied to diffusion or image autoregressive models. This sentence aims to acknowledge that dataset inference has also been studied beyond LLMs. To avoid overreach, we revised the text to clearly separate our contributions, which are specific to LLMs, from related DI work on other types of generative models. We revised Section 2 on the related work in the updated submission (please see the changes highlighted in blue).
> > > >
> > > > > **Lack significant technical contribution wrt to the closest prior work. As you report "Zhang et al. (2024a) propose to inject random and meaningless canaries into the data and then test how the LLM ranks the selected canary among all alternatives." Relative to Zhang et al. (injected random strings), your technical contribution is switching to NIDs that are naturally included in LLMs’ training sets. I would note this earlier in the paper, e.g., in the introduction too.**
> > > >
> > > > We do not merely replace random strings with NIDs. Our main contribution is **a complete post-hoc privacy auditing framework** that leverages naturally occurring identifiers in existing training data, adapts the one-run DP auditing framework to this setting (of zero-run), and makes DI practical by constructing the required in-distribution held out set from the suspect set without any retraining and distribution shift between the suspect and held out sets. We have revised the introduction to explicitly contrast Zhang et al. (2024a), who rely on injected random canaries that require modifying the training data, with our fully post-hoc approach based on naturally occurring NIDs (please see the changes shown in blue in our revised introduction).
> > > >
> > > > **References:**
> > > >
> > > > [1] "AI models collapse when trained on recursively generated data" Ilia Shumailov, Zakhar Shumaylov, Yiren Zhao, Nicolas Papernot, Ross Anderson & Yarin Gal. Nature 2024.
> > > >
> > > > [2] "Context-aware membership inference attacks against pre-trained large language models." Chang, Hongyan, Ali Shahin Shamsabadi, Kleomenis Katevas, Hamed Haddadi & Reza Shokri. In Proc. of the 2025 Conference on Empirical Methods in Natural Language Processing.
> > > >
> > > > [3] "Adaptive pre-training data detection for large language models via surprising tokens" Anqi Zhang & Chaofeng Wu. 2024.
> > > >
> > > > ---
> > > >
> > > > If the above answers address the Reviewer’s concerns, we would greatly appreciate if they could consider updating their score.

---

> > > ### Comment · Reviewer_6fad · 2025-11-25
> > > **Still missing comparison**
> > >
> > > Thanks,
> > >
> > > but still missing an empirical performance and efficiency comparison with  Zhang et al., 2024 and Zhao et al., 2025. This is important because, as I pointed out, your DI performance/gains look modest (I understand you work with more constraints, but we need baselines)

---

> > > > ### Author Response · Authors · 2025-11-30
> > > > **A comprehensive empirical performance and efficiency comparison with other methods**
> > > >
> > > > >**Thanks, but still missing an empirical performance and efficiency comparison with Zhang et al., 2024 and Zhao et al., 2025.**
> > > >
> > > > We compare our method in the pretraining setting with [2] using three Pile subsets with the Pythia 6.9B to ensure a fair and comprehensive assessment.
> > > >
> > > > We report the empirical performance, and the efficiency is compared based on the “End-to-End Execution time per data subset (in minutes)” using 4 x A100. We added the Table below to Appendix L as Table 23.
> > > >
> > > > | Subset | P-Value Zhao et al. 2025 [2] | P-Value (Ours) | End-to-End Execution time per data subset (in minutes) (Zhao et al.2025 [2])| End-to-End Execution time per data subset (in minutes)  (Ours) |
> > > > |---|---|---|---|---|
> > > > | Pile-CC | 5.64e-3 | 2.18e-34 | 1395.87 | 46.17 |
> > > > | GitHub | 8.50e-3 | 3.65e-14 | 2106.97 | 34.41 |
> > > > | Ubuntu | 4.23e-2 | 3.01e-14 | 805.22 | 21.33 |
> > > >
> > > > Our NID DI method outperforms Zhao et al.2025  [2] in both auditing effectiveness and runtime efficiency. We obtain orders of magnitude lower p-values, indicating a much stronger privacy audit than [2]. Furthermore, our method is more than an order of magnitude faster because it avoids the costly generator model training required by [2], instead relying on efficient NID evaluation via simple forward passes, making our solution far more practical.
> > > >
> > > > Moreover, we fully fine-tune the Pythia 1B model using the corresponding canaries for each method on 100 samples, as done in the other finetuning experiments. This allows us to compare all the existing dataset inference methods against each other in terms of performance (reported as p-values) and efficiency (end-to-end execution time). Note that we do the fine-tuning since the evaluation of the method from Zhang et al. 2024 [3] can be done only in this setting. Additionally, we apply the canaries proposed in Zhang et al. 2024 [3] to our dataset inference method using our statistical test (we leverage their alphabetic and hex canaries instead of our NIDs). Moreover, for [2], we still use the whole subset in the generation phase. This gives an unfair advantage to this method; however, this is a necessary step, otherwise the resulting generated held-out set would have had an even larger distribution shift. Additionally, the time reported is generation + calibration. For the generation time, we reused the time reported for the pretraining setting; therefore, the time measurement is done with 4 x A100, while all the other experiments were run using a single A100.
> > > >
> > > > In the table below, we use samples from the GitHub subset for [1,2] and ours, and the random canaries for [3].
> > > >
> > > > | **DI Method** | **P-Value Members** | **P-Value Non-Members** | **End-to-End Execution time (in minutes)** |
> > > > | :--- | :--- | :--- | :--- |
> > > > | LLM DI (Maini et al.) [1] | 9.79e-122  | 1.05e-02 | 20.43 |
> > > > | Unlock DI (Zhao et al.) [2] | 5.00e-05 | 1e00 | 2122.37 |
> > > > | Zhang et al. 2024 [3] (Hex) | 7.00e-23 | 6.70e-02 | 21.18 |
> > > > | Zhang et al. 2024 [3] (Alphabetic) | <1.00e-300 | 5.42e-01 | 20.83 |
> > > > | NID DI (Ours) | 3.31e-156 | 9.83e-01 | 21.52 |
> > > >
> > > > We observe that in the member cases, our method demonstrates strong performance. We obtain a lower p-value than Maini et al. 2024 [1] and Zhao et al. 2025 [2], and on par with Zhang et al. 2024 [3], and in the case of non-members, we obtain, similarly to other methods, a p-value close to 1.0.
> > > > Regarding efficiency, the execution time for Ours (21.52 minutes) is equally low and comparable to [1] and [3], and it is order of magnitude faster than Zhao et al. 2025 [2] (e.g., 2122.37 minutes on the GitHub subset), showcasing high computational efficiency. This represents a **99% reduction in computational cost** when comparing post-hoc auditing methods.
> > > >
> > > > Additionally, we further evaluate the methods using the whole Pile. In this setting, we cannot evaluate [2], as it requires a small distribution variability to be effective.
> > > >
> > > > | **DI Method** | **P-Value Members** | **P-Value Non-Members** | **End-to-End Execution time (in minutes)** |
> > > > | :--- | :--- | :--- | :--- |
> > > > | LLM DI (Maini et al.) [1] | 8.48e-46 | 5.85e-01 | 20.73 |
> > > > | Zhang et al. 2024 [3] (Hex) | 7.00e-23 | 6.70e-02 | 21.18 |
> > > > | Zhang et al. 2024 [3] (Alphabetic) | <1.00e-300 | 5.42e-01 | 20.83 |
> > > > | NID DI (Ours) | 4.17e-211 | 3.76e-01 | 20.67 |
> > > >
> > > > Even considering the whole subset, the trend is consistent with the previous setting.
> > > >
> > > > We added the Table above to Appendix K.7 as Tables 21 and 22.
> > > >
> > > > **References:**
> > > >
> > > > [1] Maini et al. “*LLM Dataset Inference: Did you train on my dataset?*” NeurIPS 2024.
> > > >
> > > > [2] Zhao et al. “*Unlocking Post-hoc Dataset Inference with Synthetic Data*” ICML 2025.
> > > >
> > > > [3] Zhang et al. “*Membership inference attacks cannot prove that a model was trained on your data.*” arXiv 2024 & SaTML 2025.
> > > >
> > > > [4] Meeus et al. "*Copyright Traps for Large Language Models*" ICML 2024.
> > > >
> > > > [5] Steinke et al. "Privacy Auditing with One (1) Training Run.” NeurIPS 2023.

---

> > ### Comment · Reviewer_6fad · 2025-11-25
> > **Dedicated injected canaries vs NIDs**
> >
> > Thanks, but there are many limited and unjustified choices in your new experiment. For example,
> > - Which one is the closest one to the prior work?
> > - Why are these 4 canary types chosen?
> > - Why does your limited experiment support a meaningful p-value?
> > - Why only NIDs from GitHub subset of the Pile test set? Are you sure this is representative, or it might be biased if it favours your NIDs?
> >
> > > **we generalize the method of Steinke et al. 2023 to the zero-run (post-hoc) privacy**
> >
> > This was my understanding too, but it was not clear in your manuscript.
> >
> > > **improve it by using higher cardinalities**
> >
> > Could you elaborate?

---

> > > ### Author Response · Authors · 2025-11-30
> > > **4 types of canaries, meaningful p-values, and higher cardinalities**
> > >
> > > We thank the Reviewer for the response and address all the remaining comments below.
> > >
> > > >**Which one is the closest one to the prior work?** & Why are these 4 canary types chosen?**
> > >
> > > We select these 4 types of canaries to cover a wide range of possible types both in and out of distribution. In particular, we select Alphabetic and Hex to cover random canaries (similar to the ones suggested by Zhang et al. 2024 [3], and IID to have in-distribution canaries (similar to the types of canaries used by [1,2,4]).
> > >
> > > >**Why does your experiment support a meaningful p-value?**
> > >
> > > In all the shown cases, the p-values are substantially low (<<0.001).  Moreover, the main goal is to show that NIDs are comparable to existing types of injected canaries, with the primary advantage that there is no need to add them ex-ante (before pretraining), as they are already in the training data  and allows us to generate practically unlimited held-out set.
> > >
> > > >**Why only NIDs from GitHub subset of the Pile test set? Are you sure this is representative, or it might be biased if it favours your NIDs?**
> > >
> > > We selected GitHub as a representative subset. To confirm that this choice does not bias the evaluation, we added the result for a random subset from the whole Pile. The overall trend remains consistent:
> > >
> > > | Canary Type|P-Value|
> > > |:-|-:|
> > > | Alphabetic|<1.00e-300  |
> > > | NIDs (All subsets) |4.17e-211 |
> > > | NIDs (GitHub) |3.31e-156 |
> > > | IID|4.55e-100 |
> > > | Hex|7.00e-23  |
> > >
> > > We revised Table 16 in Appendix K.1 accordingly.
> > >
> > > >**This was my understanding too, but it was not clear in your manuscript.**
> > >
> > > We would like to refer to the second contribution (lines 90-92), in which we formulate “We adapt the one-run DP auditing framework (Steinke et al., 2023) to leverage NIDs, enabling truly post-hoc DP auditing of pretrained LLMs without modifying the training process and achieving tighter lower bounds.”
> > >
> > > We also refined lines 73-74 to emphasize and clarify the distinction: “Finally, in contrast to the one training run privacy auditing by Steinke et al. 2023 [5], our method enables truly zero-run (post-hoc) audits of already pretrained LLMs.”
> > >
> > > We are happy to further revise the manuscript if a different sentence would make it even clearer.
> > >
> > > >**improve it by using higher cardinalities. Could you elaborate?**
> > >
> > > The benefit of the higher cardinality comes from the increased difficulty of the guessing task. Intuitively guessing correctly between 32 options is far more complex than between 2 options, thus providing stronger evidence of training data knowledge. Furthermore, we support this with theoretical and empirical analyses. Specifically, as we show in Appendix F, using Bernstein’s inequality, the concentration margin improves significantly as it scales with $\\sqrt\\frac{1}{mc}$. In Figure 2, we show the advantages of higher cardinality for the randomized response. Specifically, moving from $c=2$ (which is a special case of our Theorem and corresponds to the method proposed by [5]) to $c=32$ allows us to reduce the order of magnitude of samples required to show the lower bound of the $\\varepsilon$. Figure 3 further validates this, showing the empirical benefits of higher cardinalities in the zero-run auditing setting with LLMs and DP-SGD.

---

### Official Review · Reviewer_qNYy · 2025-11-12

**Soundness:** 3
**Presentation:** 3
**Contribution:** 3
**Rating:** 6
**Confidence:** 2

**Summary:**

This work aims to address two of the most significant and challenging research questions in the field of privacy auditing for LLMs: the post-hoc implementation problem of differential privacy (DP) auditing, and the dependence of dataset inference (DI) on independent and identically distributed (IID) held-out sets. The authors identify two core shortcomings of current LLM privacy auditing tools in real-world deployment: the infeasibility of DP auditing and the IID constraint in dataset inference DI. To address both issues, this work introduces a novel and elegant concept: Natural Identifiers.

**Strengths:**

1. The NID approach provides a clever, elegant, and theoretically sound solution to a long-standing and practically intractable bottleneck in the field of DI: the “IID held-out set” limitation.
2. The proposed approach seems interesting. It operates post-hoc and is capable of generating IID samples with zero distributional shift.
3. The empirical results seem promising.

**Weaknesses:**

1. The proposed method in this work relies entirely on the existence of Natural Identifiers, which are found almost exclusively in technical and code-related corpora (e.g., GitHub, StackExchange) and are virtually absent in general text corpora such as books or news articles. Therefore, this work does not appear to be a general-purpose LLM auditing tool. The authors deliberately avoid discussing this major limitation in the paper.
2. The core differential privacy claim in this paper is that it audits pre-trained LLMs; however, the differential privacy experiments seems are conducted only on fine-tuning, not on pre-training.

**Questions:**

see weakness

---

> ### Author Response · Authors · 2025-11-21
>
> We thank the Reviewer for the positive and constructive feedback. We appreciate that our privacy auditing and dataset inference methods based on NIDs are considered as *clever, elegant, and theoretically sound*.
>
> We provide detailed answers one-by-one below:
>
>
> >**The proposed method in this work relies entirely on the existence of Natural Identifiers, which are found almost exclusively in technical and code-related corpora (e.g., GitHub, StackExchange) and are virtually absent in general text corpora such as books or news articles. Therefore, this work does not appear to be a general-purpose LLM auditing tool. The authors deliberately avoid discussing this major limitation in the paper.**
>
> While we acknowledge that NIDs are more frequent in technical and code-related corpora, we detected as many as 16989 NIDs in RefinedWeb, a filtered, large-scale deduplication of CommonCrawl (please see Table 6 in Appendix D). Furthermore, discovering new task and domain-specific identifiers, such as grant numbers or trial IDs in scientific text, offers a natural path to extend NID coverage in low-NID domains.
> To validate this idea, we design task-specific NIDs for the GSM8K dataset, which consists of grade-school math word problems, and show that they enable effective DI. In the example below (for one of the samples from the GSM8K dataset), we can treat the number 48 and all derived quantities (such as 24 and 72) as a variable and substitute it with different numbers in a consistent way.
>
> Original version (suspect set, **NIDs**):
>
> **Question**
>
> Natalia sold clips to 48 of her friends in April, and then she sold half as many clips in May. How many clips did Natalia sell altogether in April and May?
>
> **Answer**
>
> Natalia sold 48/2 = <<48/2=24>>24 clips in May.
>
> Natalia sold 48+24 = <<48+24=72>>72 clips altogether in April and May.
>
> \#\#\#\# 72
>
> Generated version (held out set, **GIDs**):
>
> **Question**
>
> Natalia sold clips to 46 of her friends in April, and then she sold half as many clips in May. How many clips did Natalia sell altogether in April and May?
>
> **Answer**
>
> Natalia sold 46/2 = <<46/2=23>>23 clips in May.
>
> Natalia sold 46+23 = <<46+23=69>>69 clips altogether in April and May.
>
> \#\#\#\# 69
>
> To validate this approach, we finetune Pythia-1b on 100 NIDs based on GSM8K.
> To generate the GIDs, we prompt GPT-5.1 to first solve the task and then generate a general template with a parametric and imputable solution. Then, we generate 127 new identifiers and verify that the generated identifiers are correct by evaluating them via zero-shot prompting using GPT-5.1. The table below shows that this new type of task-specific NID can be effectively applied to DI, with all reported p-values indicating strong statistical significance for every tested number of NIDs.
>
> |Number of NIDs|P-Value |
> |--:|--:|
> | 50 | 8.43e-04 |
> | 60 | 9.56e-05 |
> | 70 | 3.35e-04 |
> | 80 |  1.63e-05 |
> | 90 |  2.12e-06 |
> | 100 | 1.60e-06 |
>
> We added into the paper the table above (Table 4) with the corresponding results and a short paragraph in Section 5 discussing them, as well as Appendix K.6 describing the experimental setup. Furthermore, we added a new Appendix L discussing the limitations of our method.
>
> >**The core differential privacy claim in this paper is that it audits pre-trained LLMs; however, the differential privacy experiments seem to be conducted only on fine-tuning, not on pre-training.**
>
> There is no publicly available DP pretrained model with a publicly known training dataset. Training a modern LLM from scratch (especially with DP) is prohibitively computationally expensive. Therefore, we finetuned the models as a proxy to show the effectiveness of our method, following the approach from [1].
>
>
> **References:**
>
> [1] "AI models collapse when trained on recursively generated data" Ilia Shumailov, Zakhar Shumaylov, Yiren Zhao, Nicolas Papernot, Ross Anderson & Yarin Gal. Nature 2024.
>
> ---
> If the above answers address the Reviewer’s concerns, we would greatly appreciate it if they could consider updating their score.

---

> > ### Author Response · Authors · 2025-11-28
> >
> > We are following up on our rebuttal and the new evidence we provided to ensure that we have fully addressed Reviewer qNYy's initial concerns.
> >
> > In particular, we have:
> > * **Expanded NID Scope**: To highlight that our method is successful beyond technical and code-related corpora, we conducted new experiments on the GSM8K dataset (detailed in Appendix K.6 and summarized in Table 4) where we show that NIDs can be generated within any dataset with structural elements. Our results demonstrate that task-specific NIDs are effective for auditing non-code domains, such as math word problems.
> > * **Clarified DP Auditing Methodology**: We provided a justification for using fine-tuning as a standard proxy for auditing pre-trained models, citing the prohibitive computational cost of DP pre-training and referencing recent relevant literature.
> >
> > If the above additions address the reviewer's concerns, we would be grateful if they could consider updating their score accordingly.

---

### Author Response · Authors · 2025-11-21

We would like to thank all the Reviewers for their valuable feedback and insightful comments, which greatly helped us further improve our submission. We thank the reviewers for noting that our work studies two important problems, namely how to carry out **zero-run** (post-hoc without (re)-training) privacy auditing (Reviewer 6fad) and how to provide the held-out set from the exact same distribution as the suspect set for dataset inference (Reviewers qNYy, TZKy). Therefore, our work introduces the *novel and elegant concept* of Natural Identifiers (Reviewer qNYy, TZKy).
We further appreciate that the reviewers find our work clever, elegant, theoretically sound, clear, well-structured, well-motivated, scalable, general, and practical (Reviewers qNYy, TZKy, uy2q, aZSB).  Finally, we thank the reviewers for recognizing our extensive empirical experiments, including privacy auditing and dataset inference, to demonstrate the effectiveness of using natural identifiers in real-world scenarios (Reviewers TZKy, uy2q, aZSB).

In the following, we present the highlights of our rebuttal:

1. **Task-specific identifiers for structured benchmarks**. Addressing the concern by multiple reviewers that NIDs might not be available in every dataset, we performed a study where we constructed different dataset-dependent NIDs. Therefore, we used the GSM8K dataset and analyzed its style. We identified possible templates and generated template-based NIDs and corresponding GIDs. Our empirical results show that these new identifiers also enable statistically significant DI on structured benchmarks. This significantly broadens the applicability of our approach beyond the NID types discussed in the original submission.

2. **NIDs vs injected and IID canaries** (Reviewer 6fad). We added a comparison of NIDs vs injected canaries for performing DI. For the canaries, we added random strings, IID text, and hexadecimal canaries (random hex digits) and fine-tuned the models. Our results show that NIDs are competitive to crafted alphabetic canaries (random letters only) and outperform IID/hexadecimal baselines while exhibiting a significant conceptual advantage over all injected canaries that they allow for *post-hoc* auditing without having to retrain/fine-tune a model.

3. **Effect of stronger MIAs on DI performance** (Reviewer 6fad). We added two recent and stronger MIAs, namely CAMIA and SURP, to our MIA features and show that stronger MIAs consistently yield lower p-values (higher power) for DI, thus highlighting that the progress in MIAs will enhance our approach.

4. **Robustness to generator misimplementation and failure modes** (Reviewer uy2q). We deliberately (as recommended) misimplement the GID generator (e.g., casing or length changes) to assess how controlled shifts between NIDs and GIDs affect DI results and to characterize the method’s robustness limits. Our results highlight that DI benefits from our approach’s ability to generate GIDs from the same distribution as NIDs.

5. **Sensitivity to NID structure and number of NIDs** (Reviewer aZSB).  We performed additional ablations where we varied NID formats and the number of identifiers. Our new results show that the longer and more structured identifiers yield better results, whereas shorter formats still return highly significant results. Furthermore, as expected, increasing the number of NIDs improves DI's statistical power.

6. **Multi-seed DP lower-bound estimates and confidence intervals** (Reviewer aZSB). We repeated DP auditing over multiple seeds and report the mean and standard deviation of $\varepsilon$, confirming low variance and the benefits of higher cardinality.


We believe these revisions meaningfully address the Reviewers' concerns and enhance the practical relevance of our work. We sincerely appreciate the Reviewers’ thoughtful and constructive feedback. Please let us know if there is any additional information we can provide to support your evaluation.

---

### Meta-Review · Area_Chair_r7c6 · 2026-01-04

**Summary:**

The authors propose using natural identifiers (NIDs) such as hashes and unique IDs that frequently appear in large text corpora. These identifiers define a subset of the training data whose structure allows the generation of unlimited IID variants that are guaranteed not to appear in training. This enables two key auditing tasks. First, for differential privacy audits, NIDs act as realistic substitutes for hand-crafted canaries. Second, for dataset inference, the generated identifiers serve as a true held-out negative set from the same distribution as the suspect dataset. The approach provides a practical, post-hoc way to audit pretrained language models without retraining or modifying the training pipeline.

**Reviewer Concerns:**

### 1. Reliance on the presence of NIDs in training corpora [qNYy, TZKy, uy2q]

[Reviewers]
- NIDs are most common in technical corpora; may be scarce in books, news, or specialised domains [qNYy, TZKy].
- NID-based audits are not robust to potential adversarial pre-processing; a data curator could strip NIDs with simple regex filters [uy2q].
- Model developers could easily remove NIDs to avoid auditing [TZKy].

[Authors] NIDs are present even in curated corpora such as RefinedWeb and Dolma (with quantitative results). Auditors need only ~100 NIDs, whereas removing all NID forms across trillions of tokens would be costly for the model trainer. Introduce task-specific NIDs and demonstrate this on GSM8K. Show that dataset inference remains statistically significant.

[AC] The fundamental/conceptual concern remains, but should not be a blocker against acceptance.


### 2. Compare against prior approaches not using NIDs [6fad]

[Reviewer] For DI, request comparisons against Zhang et al. (2024/2025).

[Author] Experiments comparing canaries vs NIDs; NIDs are competitive and outperform IID/hex (albeit with the post-hoc advantage); Included stronger MIA variants (CAMIA, SURP).

[Reviewer] More justification of design choices and representativeness needed

[Author] Extended tables and clarified scope.

[AC] The conversation is not fully resolved. However, I would not consider this a killer argument against the paper.



### 3. It's fiinetuning, not pretraining [qNYy]

[Reviewer] Paper claims to audit pretrained models, but DP experiments are done on finetuning rather than full pretraining.

[Author] Cite practical constraints; no public DP-pretrained models with known data; full DP pre-training is prohibitively expensive; Finetuning is a standard proxy used in prior work.

[AC] Author's position is reasonable.


### 4. Sensitivity to generator failures [uy2q]

[Reviewer] Subtle mismatches between NIDs and generated GIDs could leak membership signal?

[Author] Stress-tests with mis-implemented generators. Some mismatches do cause false positives.

[Follow-up] uy2q confirmed their initial score to accept.

[AC] Concern is reasonable, but would not kill the paper.

**Reviewer Scores:**

qNYy: 6 --> 6 (expected to remain convinced about the paper)
6fad: 2 --> 4 (concerns addressed at least to some degree)
TZKy: 6 --> 8 (explicit)
uy2q: 8 --> 8 (explicit)
aZSB: 8 --> 8 (expected to retain score)

---

### Decision · Program_Chairs · 2026-01-26

Accept (Poster)